# A bidirectional corticoamygdala circuit for the encoding and retrieval of detailed reward memories

Ana C Sias[1], Ashleigh K Morse[1], Sherry Wang[1], Venuz Y Greenfield[1], Caitlin M Goodpaster[1], Tyler M Wrenn[1], Andrew M Wikenheiser[1,2,3], Sandra M Holley[4], Carlos Cepeda[4], Michael S Levine[2,4], Kate M Wassum[1,2,3,5]*

[1]Department of Psychology, University of California, Los Angeles, Los Angeles, United States; [2]Brain Research Institute, University of California, Los Angeles, Los Angeles, United States; [3]Integrative Center for Learning and Memory, University of California, Los Angeles, Los Angeles, United States; [4]Intellectual and Developmental Disabilities Research Center, Semel Institute for Neuroscience and Human Behavior, David Geffen School of Medicine, University of California, Los Angeles, Los Angeles, United States; [5]Integrative Center for Addictive Disorders, University of California, Los Angeles, Los Angeles, United States

**Abstract** Adaptive reward-related decision making often requires accurate and detailed representation of potential available rewards. Environmental reward-predictive stimuli can facilitate these representations, allowing one to infer which specific rewards might be available and choose accordingly. This process relies on encoded relationships between the cues and the sensory-specific details of the rewards they predict. Here, we interrogated the function of the basolateral amygdala (BLA) and its interaction with the lateral orbitofrontal cortex (lOFC) in the ability to learn such stimulus-outcome associations and use these memories to guide decision making. Using optical recording and inhibition approaches, Pavlovian cue-reward conditioning, and the outcome-selective Pavlovian-to-instrumental transfer (PIT) test in male rats, we found that the BLA is robustly activated at the time of stimulus-outcome learning and that this activity is necessary for sensory-specific stimulus-outcome memories to be encoded, so they can subsequently influence reward choices. Direct input from the lOFC was found to support the BLA in this function. Based on prior work, activity in BLA projections back to the lOFC was known to support the use of stimulus-outcome memories to influence decision making. By multiplexing optogenetic and chemogenetic inhibition we performed a serial circuit disconnection and found that the lOFC→BLA and BLA→lOFC pathways form a functional circuit regulating the encoding (lOFC→BLA) and subsequent use (BLA→lOFC) of the stimulus-dependent, sensory-specific reward memories that are critical for adaptive, appetitive decision making.

*For correspondence:
kwassum@ucla.edu

## Introduction

To make good decisions we must accurately anticipate the potential outcomes (e.g. rewarding events) that might be available in our current situation, or state. When not readily observable, we can infer the availability of these outcomes from predictive environmental stimuli (e.g. restaurant logos on a food-delivery app). Pavlovian *stimulus-outcome associative memories* enable such cues to trigger representations of their associated outcomes, thus facilitating the state-dependent outcome expectations that influence decision making (*Balleine and Dickinson, 1998*; *Delamater, 2012*; *Fanselow and Wassum, 2015*). Often our decisions require detailed information about the available outcomes (e.g. flavor, nutritional content, texture). For example, when deciding between items of

similar valence (e.g. to have pizza or sushi for dinner). To enable such decisions, stimulus-outcome memories can be quite rich, including the sensory-specific, identifying details of the predicted reward (*Delamater and Oakeshott, 2007*; *Fanselow and Wassum, 2015*). Failure to properly encode or use such memories can lead to poor reward-related choices, a hallmark feature of myriad psychiatric diseases. Yet much is unknown of the neural circuits that support stimulus-outcome memory.

One potential hub for stimulus-outcome memory is the basolateral amygdala (BLA) (*Wassum and Izquierdo, 2015*). Long known for its function in emotional learning, the BLA is thought to link predictive stimuli with valence and to relay that valence for adaptive behavior (e.g. approach/avoidance) (*Baxter and Murray, 2002*; *Janak and Tye, 2015*; *Pignatelli and Beyeler, 2019*; *Tye, 2018*). But the BLA does more than valence. Mounting evidence, primarily collected with lesion and inactivation strategies, suggests the BLA mediates appetitive behaviors that require a rich sensory-specific representation of the expected reward. For example, the BLA is needed for reward-predictive cues to bias choice between two distinct rewards (*Blundell et al., 2001*; *Corbit and Balleine, 2005*; *Hatfield et al., 1996*; *Ostlund and Balleine, 2008*). Although the BLA's function in the *expression* of such behaviors has been established, temporal limitations of BLA lesions preclude interpretations of BLA function in stimulus-outcome *learning*. The BLA is known to be essential for the learning of cued fear (*Muller et al., 1997*; *Sengupta et al., 2018*), but behavioral limitations of these studies preclude understanding of whether the BLA is involved in encoding the sensory-specific details of the outcome. Thus, it remains unknown whether the BLA is involved in encoding the sensory-specific stimulus-outcome memories that enable adaptive choices, or if the BLA primarily functions to assign general valence to a cue. Moreover, little is known of the endogenous activity or circuit function underlying any potential role for the BLA in the formation of appetitive stimulus-outcome memories.

To address these gaps in knowledge, here we used optical recording and inhibition approaches in male rats to examine the BLA's function in the encoding of stimulus-outcome memories for two unique food rewards. To assess the extent of stimulus-outcome memory encoding, we used the outcome-selective Pavlovian-to-instrumental transfer (PIT) test to measure the ability of a reward-paired stimulus to trigger a sensory-specific representation of its predicted reward and thus bias reward-seeking choice (*Colwill and Motzkin, 1994*; *Corbit and Balleine, 2016*; *Gilroy et al., 2014*; *Kruse et al., 1983*).

## Results

### BLA neurons respond to rewards and cues during appetitive Pavlovian stimulus-outcome learning

We first asked whether and when the BLA is active during the encoding of stimulus-outcome memories (*Figure 1a*). To condition cues that set the 'state' for a the availability of a specific reward and engender a sensory-specific representation of that reward, we used a two food outcome Pavlovian conditioning task. Each of 2, 2 min auditory conditional stimuli (CSs; white noise and tone) were associated with intermittent delivery of 1 of 2 distinct food rewards (sucrose solution or food pellets; e.g. white noise-sucrose/tone-pellet). This conditioning has been shown to engender the encoding of detailed, sensory-specific stimulus-outcome memories as measured by the cue's ability to subsequently promote instrumental choice for the specific predicted reward during a PIT test (*Lichtenberg et al., 2017*; *Lichtenberg and Wassum, 2017*; *Malvaez et al., 2015*; *Ostlund and Balleine, 2008*), as well as the sensitivity of the conditional food-port approach response to sensory-specific devaluation of the predicted reward (*Lichtenberg et al., 2017*) or degradation of the stimulus-outcome contingency (*Ostlund and Balleine, 2008*). Food-deprived, male rats (*N* = 11) received 8 Pavlovian conditioning sessions. During each session, each cue was presented four times (variable intertrial interval, average = 3 min) for 2 min, during which its associated reward was intermittently delivered on average every 30 s. Rats demonstrated simple Pavlovian conditioning by gradually increasing their goal approach responses (entries into the food-delivery port) during the cue probe periods (after cue onset, before reward delivery) across training (*Figure 1h*; Training: $F_{(2.4,24.3)}$ = 13.18, p<0.0001; see also *Figure 1—figure supplement 1*).

To characterize the endogenous activity of BLA neurons during the encoding of appetitive stimulus-outcome memories, we used fiber photometry to image the fluorescent activity of the genetically

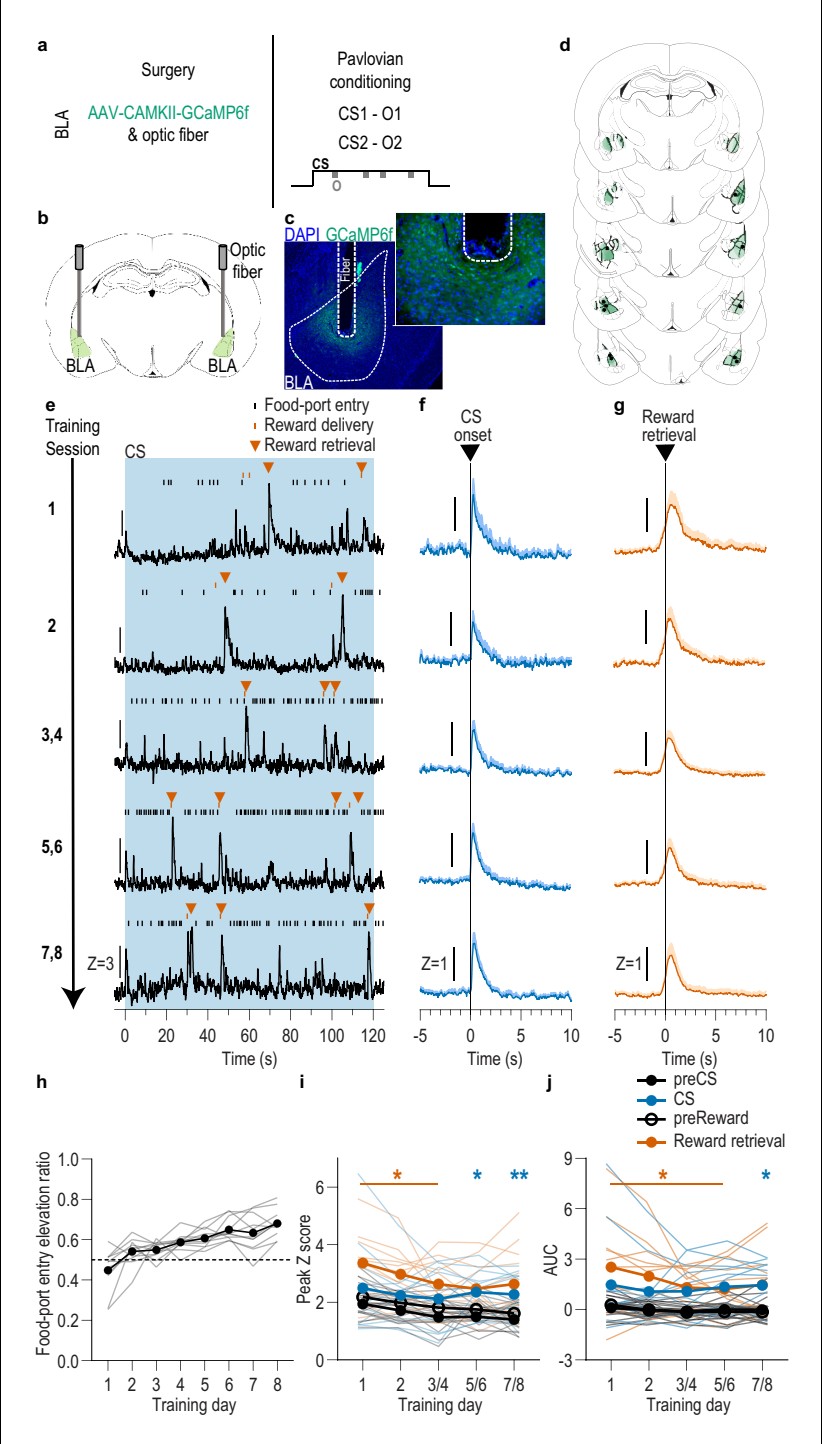

**Figure 1.** BLA neurons are activated during stimulus-outcome learning. (a). Procedure schematic. CS, conditional stimulus (white noise or tone); O, outcome (sucrose solution or food pellet). (b) Schematic of fiber photometry approach for imaging bulk calcium activity in BLA neurons. (c) Representative fluorescent image of GCaMP6f expression and fiber placement in the BLA. (d) Schematic representation of GCaMP6f expression and placement of optical fiber tips in BLA for all subjects. Brain slides from *Paxinos and Watson, 1998*. (e) Representative examples of GCaMP6f fluorescence changes (Z-scored ΔF/F) in response to CS presentation (blue box), reward delivery, and reward retrieval (first food-port entry following reward delivery) across days of training. Traces from the last 6 days of training were selected from one of each two-session bin. See *Figure 1—figure supplement 2* for raw GCaMP and isosbestic signal fluctuations. (f-g) Trial-averaged GCaMP6f fluorescence changes (Z-scored

*Figure 1 continued on next page*

*Figure 1 continued*

ΔF/F) in response to CS onset (f; blue) or reward retrieval during the CS (g; orange) across days of training. Shading reflects between-subjects s.e.m. Data from the last six sessions were averaged across two-session bins (3/4, 5/6, and 7/8). (h) Elevation [(CS probe entry rate)/(CS probe entry rate + preCS entry rate)] in food-port entries during the CS probe period (after CS onset, before first reward delivery), averaged across trials and across the 2 CSs for each day of Pavlovian conditioning. Gray lines represent individual subjects. (i-j) Trial-averaged quantification of maximal (i; peak) and area under the GCaMP Z-scored ΔF/F curve (j; AUC) during the 3 s period following CS onset or reward retrieval compared to equivalent baseline periods immediately prior to each event. Thin light lines represent individual subjects. $N = 11$ (see *Figure 1—figure supplement 3* for data from $N = 8$ subjects with longitudinal data from each session). *p<0.05, **p<0.01 relative to pre-event baseline. See *Figure 1—source data 1*.

The online version of this article includes the following source data and figure supplement(s) for figure 1:

**Source data 1.** Source data for *Figure 1* and *Figure 1—figure supplement 1–5*.
**Figure supplement 1.** Food-port entry rate during Pavlovian conditioning for BLA fiber photometry GCaMP6f imaging experiment.
**Figure supplement 2.** Representative examples of raw GCaMP6f and isosbestic fluorescent changes in response to cue presentation and reward delivery and retrieval across days of training.
**Figure supplement 3.** BLA neurons are activated during stimulus-outcome learning across each of the eight Pavlovian conditioning sessions.
**Figure supplement 4.** BLA reward responses aligned to reward delivery during Pavlovian conditioning.
**Figure supplement 5.** Food-port entries during the CS in the absence of reward do not trigger a BLA response.

encoded calcium indicator GCaMP6f (*Chen et al., 2013*) each day during Pavlovian conditioning (*Figure 1b–d*). GCaMP6f was expressed preferentially in principal neurons based on expression of calcium/calmodulin-dependent protein kinase, CaMKII (*Butler et al., 2011*; *Tye et al., 2011*). Data from the eight training sessions were binned into five conditioning phases, session 1, session 2, sessions 3/4, 5/6, and 7/8. Thus, data from the last six sessions were averaged across two-session bins. As can be seen in the representative examples (*Figure 1e*; see also *Figure 1—figure supplement 2*), or group-averaged traces (*Figure 1f–g*), BLA neurons were robustly activated by both cue onset and reward retrieval (first food-port entry after reward delivery) throughout Pavlovian conditioning. Across training, both the cues and rewards caused a similar elevation in the peak calcium response (*Figure 1i*; Event v. baseline: $F_{(0.4,3.9)} = 36.02$, p=0.007; Training: $F_{(2.8,28.1)} = 4.29$, p=0.01; Event type (CS/US) and interactions between factors, lowest p=0.18) and area under the calcium curve (AUC; *Figure 1j*; Event v. baseline: $F_{(0.3,3.4)} = 35.23$, p=0.01, Training, Event type, and interactions between factors, lowest p=0.23; see also *Figures 1–3*). Analysis of each event relative to its immediately preceding baseline period confirmed that BLA neurons were robustly activated by CS onset as reflected in the peak calcium response (CS: $F_{(1,10)} = 7.25$, p=0.02; Training: $F_{(2.5, 24.5)}=1.88$, p=0.17; CS x Training: $F_{(1.2, 12.4)}=0.54$, p=0.51) and AUC (CS: $F_{(1,10)} = 6.28$, p=0.03; Training: $F_{(1.9,19.3)} = 0.40$, p=0.67; CS x Training: $F_{(1.2,11.7)} = 0.17$, p=0.73), as well as at reward retrieval during the cue [(Peak, Reward: $F_{(1,10)} = 16.82$, p=0.002; Training: $F_{(1.9,19.4)} = 3.41$, p=0.06; Reward x Training: $F_{(1.7,16.8)} = 0.88$, p=0.42) (AUC, Reward: $F_{(1,10)} = 15.21$, p=0.003; Training: $F_{(1.6,15.7)} = 2.13$, p=0.16; Reward x Training: $F_{(1.5,14.8)} = 1.25$, p=0.30)]. The same BLA reward response could also be detected when the data were aligned to reward delivery (*Figure 1—figure supplement 4*). There were no significant BLA activity changes detected in response to food-port entries absent reward (*Figure 1—figure supplement 5*), indicating that reward retrieval responses resulted from reward experience rather than the act of entering the food port. Thus, BLA neurons are active at the most critical time for the encoding of stimulus-outcome memories, when the reward is experienced during the cue (i.e. the stimulus-outcome pairing).

It was surprising that responses to the cues were present on the first conditioning session, particularly in light of evidence that BLA responses to both appetitive and aversive cues increase across learning (*Crouse et al., 2020*; *Johansen et al., 2010*; *Lutas et al., 2019*; *Tye et al., 2008*). This could reflect a non-associative, novelty response to either or both the tone or white noise presentation. To examine this and, thus, evaluate whether the BLA cue responses later in training were due to stimulus-outcome learning, we repeated the experiment in a separate group of naive rats, but this time omitted the reward delivery during conditioning (*Figure 2a–c*; $N = 6$). Instead, the rewards were delivered unpaired with the cues several hours after each session in a distinct context. Like

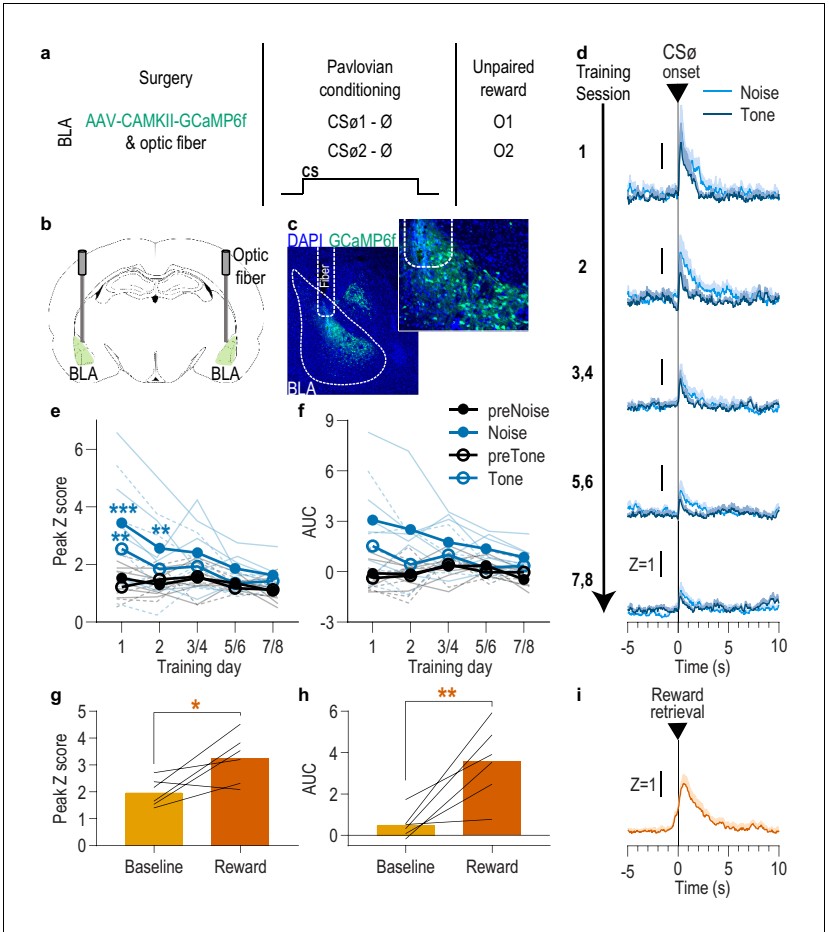

**Figure 2.** BLA neurons are only transiently activated by stimuli if they are not paired with reward. (**a**). Procedure schematic. $CS_\varnothing$, neutral stimulus; $\varnothing$, no reward outcome; O, outcome (sucrose solution or food pellet). (**b**) Schematic of fiber photometry approach for imaging bulk calcium activity in BLA neurons. (**c**) Representative fluorescent image of GCaMP6f expression and fiber placement in the BLA. (**d**) Trial-averaged GCaMP6f fluorescence change (Z-scored ΔF/F) in response to noise and tone $CS_\varnothing$ onset across days. Shading reflects between-subjects s.e.m. (**e-f**) Trial-averaged quantification of maximal (e; peak) and area under the GCaMP Z-scored ΔF/F curve (f; AUC) during the 3 s following noise and tone $CS_\varnothing$ onset compared to equivalent baseline periods immediately prior to each event. Thin light lines represent individual subjects (solid = Noise, dashed = Tone). (**g-h**) Trial-averaged quantification of maximal (g; peak) and area under the GCaMP Z-scored ΔF/F curve (h; AUC) during the 3 s following retrieval of the unpaired reward compared to equivalent baseline period immediately prior reward retrieval. Lines represent individual subjects. (**i**) Trial-averaged GCaMP6f fluorescence (Z-scored ΔF/F) in response to unpaired reward, averaged across reward type. Shading reflects between-subjects s. e.m. $N = 6$. *p<0.05, **p<0.01 relative to pre-event baseline. See *Figure 2—source data 1*.

The online version of this article includes the following source data for figure 2:

**Source data 1.** Source data for *Figure 2*.

presentation of the reward-predictive cues, presentation of either the tone or white noise stimulus unpaired with reward ($CS_\varnothing$) robustly activated BLA neurons during the first session, but, in contrast to the reward-predictive cues, this effect habituated over sessions (*Figure 2d*). Both tone and noise elicited a similar elevation in the peak calcium response that was largest on session one and diminished with subsequent days of exposure (*Figure 2e*; Session x $CS_\varnothing$: $F_{(4,20)} = 3.25$, p=0.03; $CS_\varnothing$ presence: $F_{(0.4,2.1)} = 4.84$, p=0.13; $CS_\varnothing$ type (white noise v. tone): $F_{(0.3,1.5)} = 7.03$, p=0.12; Session: $F_{(2.3,11.7)} = 3.27$, p=0.07; Session x $CS_\varnothing$ type: $F_{(4,20)} = 1.42$, p=0.26; $CS_\varnothing$ x $CS_\varnothing$ type: $F_{(0.5,2.3)} = 9.69$, p=0.07; Session x $CS_\varnothing$ x $CS_\varnothing$ type: $F_{(0.6,3.2)} = 0.80$, p=0.37). The effect was similar when quantified using area under the calcium curve (*Figure 2f*; Session x $CS_\varnothing$: $F_{(4,20)} = 2.65$, p=0.06; $CS_\varnothing$ presence: $F_{(0.5,2.4)} = 5.07$, p=0.12; $CS_\varnothing$ type: $F_{(0.3,1.4)} = 4.81$, p=0.14; Session: $F_{(2.6,12.8)} = 1.55$, p=0.25; Session

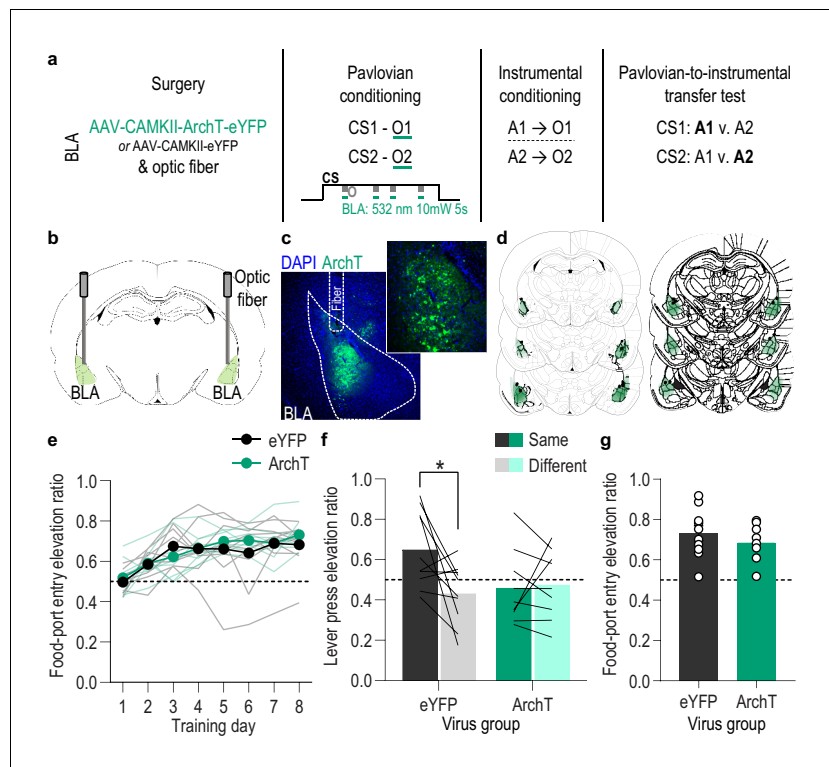

**Figure 3.** Optical inhibition of BLA neurons during stimulus-outcome pairing attenuates the encoding of stimulus-outcome memories. (**a**). Procedure schematic. CS, conditional stimulus (white noise or tone); O, outcome (sucrose solution or food pellet); A, action (left or right lever press). (**b**) Schematic of optogenetic strategy for bilateral inhibition of BLA neurons. (**c**) Representative fluorescent image of ArchT-eYFP expression and fiber placement in the BLA. (**d**) Schematic representation of ArchT-eYFP expression and placement of optical fiber tips in BLA for all subjects. (**e**) Elevation [(CS probe entry rate)/(CS probe entry rate + preCS entry rate)] in food-port entries during the CS probe period (after CS onset, before first reward delivery), averaged across trials and CSs for each day of Pavlovian conditioning. Thin light lines represent individual subjects. (**f**) Elevation in lever presses on the lever that earned the same outcome as the presented CS (Same; [(presses on Same lever during CS)/(presses on Same lever during CS + Same presses during preCS)], averaged across trials and across CSs), relative to the elevation in responding on the alternate lever (Different; [(presses on Different lever during CS)/(presses on Different lever during CS + Different presses during preCS)], averaged across trials and across CSs) during the PIT test. Lines represent individual subjects. (**g**) Elevation in food-port entries to CS presentation (averaged across trials and CSs) during the PIT test. Circles represent individual subjects. ArchT, *N* = 9; eYFP, *N* = 10. *p<0.05. See *Figure 3—source data 1*.

The online version of this article includes the following source data and figure supplement(s) for figure 3:

**Source data 1.** Source data for *Figure 3* and *Figure 3—figure supplement 1–3*.
**Figure supplement 1.** Green light activation of ArchT hyperpolarizes and attenuates the firing of BLA cells.
**Figure supplement 2.** Food-port entry and press rates during Pavlovian conditioning and PIT test for BLA optical inhibition experiment.
**Figure supplement 3.** Inhibition of BLA neurons unpaired with reward delivery does not disrupt the encoding of stimulus-outcome memories.

x $CS_\varnothing$ type: $F_{(4,20)} = 1.14$, p=0.37; $CS_\varnothing$ x $CS_\varnothing$ type: $F_{(0.5,2.4)} = 10.43$, p=0.06; Session x $CS_\varnothing$ x $CS_\varnothing$ type: $F_{(0.7,3.7)} = 1.81$, p=0.24). To check whether the decline of the $CS_\varnothing$ response was due simply to signal degradation over time, following the last $CS_\varnothing$ session we recorded BLA calcium responses to unpredicted reward delivery. Rewards were capable of robustly activating the BLA (*Figure 2g–i*; Peak, $t_5 = 2.93$, p=0.03; AUC, $t_5 = 4.07$, p=0.01). This positive control indicates that the decline of the BLA $CS_\varnothing$ response was due to stimulus habituation, not signal degradation. Thus, the BLA response to cue presentation during early Pavlovian conditioning likely reflects a non-associative novelty effect that habituates with subsequent exposure, indicating that the BLA responses to the reward-predictive cues later in training (*Figure 1*) largely result from the association with reward.

## BLA neuron activity is necessary during outcome experience to encode appetitive Pavlovian stimulus-outcome memories

We found that BLA neurons are robustly activated at the time at which stimulus-reward memories can be formed: when the reward is experienced during a predictive cue. We next asked whether this activity is necessary for such learning and, if so, whether it is necessary for encoding sensory-specific stimulus-outcome memories (*Figure 3a*). We expressed the inhibitory opsin archaerhodopsin T (ArchT; *N* = 9) or eYFP control (*N* = 10) in BLA, primarily, principal neurons (*Figure 3b–d*) to allow green light (532 nm,~10 mW) to transiently hyperpolarize and inhibit the activity of these cells (*Figure 3—figure supplement 1*). Rats were again given 8 Pavlovian conditioning sessions during which each of 2 distinct, 2 min auditory CSs was paired with intermittent delivery of one specific food reward (8 of each CS/session). During each Pavlovian conditioning session, we optically inhibited the activity of BLA neurons during each cue. We restricted inhibition to 5 s concurrent with the delivery and retrieval of each food reward because this is the time at which the stimulus-outcome pairing occurs and when we found the BLA to be endogenously active (*Figure 1*). Optical inhibition of BLA neurons at reward experience during Pavlovian conditioning did not impede the development of the Pavlovian conditional goal-approach response to the cue sampled prior to reward delivery (*Figure 3e*; Training: $F_{(3.8,64.9)}$ = 17.53, p<0.0001; Virus (eYFP v. ArchT): $F_{(1,17)}$ = 0.19, p=0.67; Virus x Training: $F_{(7,119)}$ = 1.28, p=0.26; see also *Figure 3—figure supplement 2a*). This general conditional response at the shared food port, however, does not require that the subjects have learned the sensory-specific details of the predicted reward. To test for such stimulus-outcome memory encoding, we gave subjects instrumental conditioning followed by a PIT test. Both were conducted without any manipulation. During instrumental conditioning, rats were trained that two different actions (left or right lever press) each earned one of the unique food rewards (e.g. left press→sucrose/right press→pellets; *Figure 3—figure supplement 2b*). At the PIT test both levers were present, but lever pressing was not rewarded. Each CS was presented four times (also without accompanying reward), with intervening CS-free baseline periods, to assess its influence on action performance and selection in the novel choice scenario. Because the cues are never associated with the instrumental actions, this test assesses the ability to, upon cue presentation, retrieve a memory of the specific predicted reward and use it to motivate choice of the action known to earn the same unique reward (*Colwill and Motzkin, 1994*; *Corbit and Balleine, 2016*; *Gilroy et al., 2014*; *Kruse et al., 1983*). If subjects had encoded detailed stimulus-outcome memories during Pavlovian conditioning, then the CS should cause them to increase presses selectively on the lever that, during training, earned the *same* outcome as predicted by that cue. Controls showed this outcome-specific PIT effect (*Figure 3f*). Conversely, the cues were not capable of influencing lever-press choice in the group for which the BLA was inhibited at the time of outcome experience during Pavlovian conditioning (*Figure 3f*; Virus x Lever: $F_{(1,17)}$ = 5.10, p=0.04; Virus: $F_{(1,17)}$ = 1.41, p=0.25; Lever (Same v. Different): $F_{(1,17)}$ = 3.84, p=0.07; see also *Figure 3—figure supplement 2c*). As in training, during this PIT test the conditional goal-approach response was similar between groups (*Figure 3g*; $t_{17}$ = 0.94, p=0.36; see also *Figure 3—figure supplement 2d*). Thus, BLA neuronal activity is not needed for the learning that supports general conditional approach responses, but is necessary, specifically at the time of outcome experience, to link the sensory-specific details of the outcome to a predictive cue. Such encoding is critical for that cue to subsequently guide decision making.

An alternative possibility is that the total amount of inhibition compromised BLA activity more broadly. That is, that BLA activity per se rather than specifically at the time of stimulus-outcome pairing mediates the encoding of stimulus-outcome memories. To rule this out, we repeated the experiment in a new cohort of naive rats in which we matched the frequency and duration of inhibition to the experimental group, but delivered it during baseline pre-CS periods during Pavlovian conditioning. This inhibition had no effect on the subsequent influence of the cues on instrumental choice behavior during the PIT test (*Figure 3—figure supplement 3*), confirming that BLA activity specifically at the time of stimulus-outcome pairing mediates the encoding of detailed stimulus-outcome memories.

## lOFC→BLA projections are necessary for encoding Pavlovian stimulus-outcome memories

We found that activity in BLA neurons at the time of reward delivery/experience mediates encoding of the relationship between that specific rewarding event and the environmental stimulus that predicts it. We next asked which BLA input might facilitate this function. The orbitofrontal cortex (OFC) is a prime candidate. The OFC sends dense glutamatergic innervation to the BLA (*Aggleton et al., 1980*; *Carmichael and Price, 1995*; *Heilbronner et al., 2016*; *Lichtenberg et al., 2017*; *Malvaez et al., 2019*; *Price, 2007*) and is itself implicated in appetitive learning (*Baltz et al., 2018*; *Murray and Izquierdo, 2007*; *Ostlund and Balleine, 2007b*; *Rudebeck and Rich, 2018*). BLA inputs from the lateral (lOFC), rather than medial OFC subregion, have previously been shown to be involved in learning information about a reward (i.e. its incentive value) (*Malvaez et al., 2019*), but are not required for retrieving appetitive memories (*Lichtenberg et al., 2017*; *Malvaez et al., 2019*). Thus, this pathway might play a critical role specifically in *forming* stimulus-outcome associative memories. To evaluate this, we used pathway-specific optical inhibition to ask whether activity in lOFC→BLA projections mediates the encoding of stimulus-outcome memories (*Figure 4a*). We expressed ArchT (*N* = 8) or eYFP control (*N* = 8) in lOFC neurons and detected expression in lOFC axons and terminals in the BLA in the vicinity of implanted optical fibers (*Figure 4b–d*). Green light (532 nm, ~10 mW) was used to inhibit lOFC axons and terminals in the BLA (*Figure 4—figure supplement 1*). Subjects received Pavlovian conditioning, as above, and inhibition was again restricted to 5 s during the delivery and retrieval of each reward during each cue. Similar to inhibition of BLA neurons, optical inhibition of lOFC→BLA projection activity during stimulus-outcome pairing did not affect the development of the Pavlovian conditional goal-approach response (*Figure 4e*; Training: $F_{(3.9,54.3)}$ = 7.84, p<0.0001; Virus: $F_{(1,14)}$ = 0.22, p=0.65; Virus x Training: $F_{(7,98)}$ = 0.43, p=0.88; see also *Figure 4—figure supplement 2a*) or its expression during the PIT test (*Figure 4g*; $t_{14}$ = 0.49, p=0.63; see also *Figure 4—figure supplement 2d*). It did, however, attenuate encoding of sensory-specific stimulus-outcome memories as evidenced by the subjects' inability to later use those memories to allow cue presentation to bias choice behavior during the PIT test (*Figure 4f*; Virus x Lever: $F_{(1,14)}$ = 6.49, p=0.02; Virus: $F_{(1,14)}$ = 0.04, p=0.85; Lever: $F_{(1,14)}$ = 7.10, p=0.02; see also *Figure 4—figure supplement 2c*). Thus, activity in lOFC→BLA projections regulates the encoding of detailed, sensory-specific stimulus-outcome memories. Together, with prior evidence that inactivation of lOFC→BLA projections does not disrupt the *expression* of outcome-selective PIT (*Lichtenberg et al., 2017*), these data suggest that activity in lOFC→BLA projections mediates the encoding, but not retrieval of stimulus-outcome memories.

## lOFC→BLA→lOFC is a stimulus-outcome memory circuit

Collectively, the data show that the BLA, with help from direct lOFC input, mediates the encoding of the detailed cue-reward memories that enable the cues to trigger the sensory-specific reward outcome representations that influence decision making. The lOFC-BLA circuit is bidirectional. The BLA sends dense excitatory projections back to the lOFC (*Barreiros et al., 2021*; *Lichtenberg et al., 2017*; *Morecraft et al., 1992*). Activity in these projections mediates the representation of expected outcomes in the lOFC (*Rudebeck et al., 2013*; *Rudebeck et al., 2017*; *Schoenbaum et al., 2003*) and the use of stimulus-outcome memories to guide choice (*Lichtenberg et al., 2017*). But it remains unknown whether BLA→lOFC projection activity enables the use of the associative information that is learned via activation of lOFC→BLA projections. That is, whether lOFC→BLA→lOFC is a functional stimulus-outcome memory encoding and retrieval circuit or whether lOFC→BLA and BLA→lOFC projections tap in to independent, parallel information streams. Indeed, stimulus-outcome memories are highly complex including multifaceted information about outcome attributes (e.g. value, taste, texture, nutritional content, category, probability, timing, etc.) and related consummatory and appetitive responses (*Delamater and Oakeshott, 2007*). Therefore, we next asked whether the lOFC→BLA and BLA→lOFC pathways form a functional stimulus-outcome memory encoding and retrieval circuit, that is, whether the sensory-specific associative information that requires lOFC→BLA projections to be encoded also requires activation of BLA→lOFC projections to be used to guide decision making, or whether these are independent, parallel pathways, tapping into essential but independent streams of information.

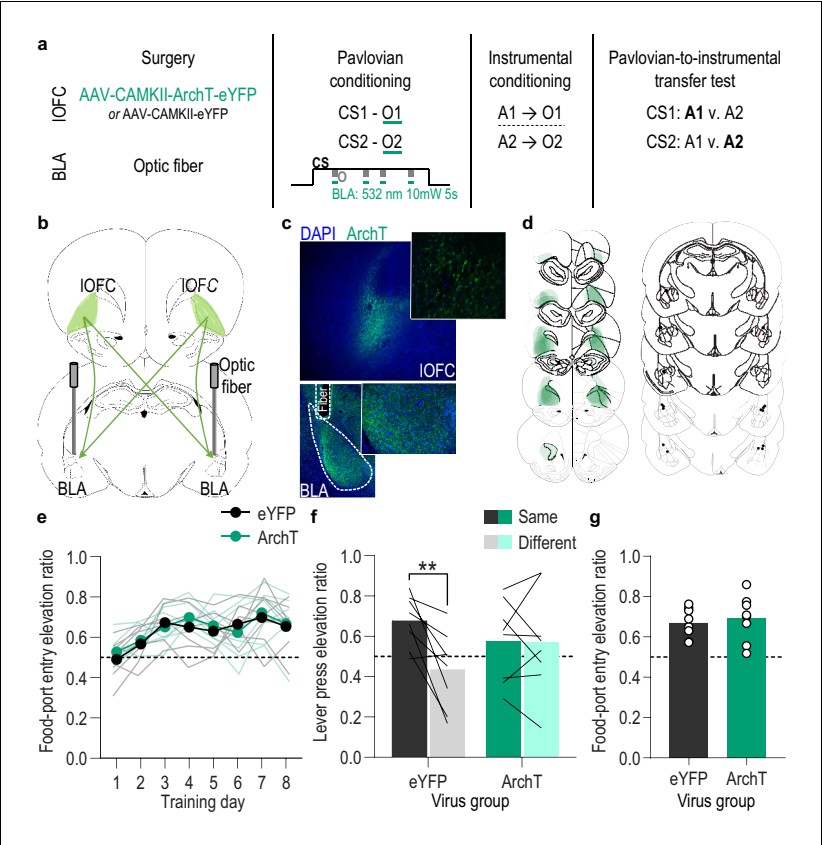

**Figure 4.** Optical inhibition of lOFC terminals in the BLA during stimulus-outcome pairing attenuates the encoding of stimulus-outcome memories. (**a**). Procedure schematic. CS, conditional stimulus (white noise or tone); O, outcome (sucrose solution or food pellet); A, action (left or right lever press). (**b**) Schematic of optogenetic strategy for bilateral inhibition of lOFC axons and terminals in the BLA. (**c**) Top: Representative fluorescent image of ArchT-eYFP expression in lOFC cell bodies. Bottom: Representative image of fiber placement in the vicinity of immunofluorescent ArchT-eYFP-expressing lOFC axons and terminals in the BLA. (**d**) Schematic representation of ArchT-eYFP expression in lOFC and placement of optical fiber tips in BLA for all subjects. (**e**) Elevation [(CS probe entry rate)/(CS probe entry rate + preCS entry rate)] in food-port entries during the CS probe period (after CS onset, before first reward delivery), averaged across trials and CSs for each day of Pavlovian conditioning. Thin light lines represent individual subjects. (**f**) Elevation in lever presses on the lever that earned the same outcome as the presented CS (Same; [(presses on Same lever during CS)/(presses on Same lever during CS + Same presses during preCS)], averaged across trials and across CSs), relative to the elevation in responding on the alternate lever (Different; [(presses on Different lever during CS)/(presses on Different lever during CS + Different presses during preCS)], averaged across trials and across CSs) during the PIT test. Lines represent individual subjects. (**g**) Elevation in food-port entries to CS presentation (averaged across trials and CSs) during the PIT test. Circles represent individual subjects. ArchT, *N* = 8; eYFP, *N* = 8. **p<0.01. See *Figure 4—source data 1*.
The online version of this article includes the following source data and figure supplement(s) for figure 4:

**Source data 1.** Source data for *Figure 4* and *Figure 4—figure supplement 1–2*.
**Figure supplement 1.** Green light activation of ArchT-expressing lOFC terminals reduces spontaneous activity in BLA neurons.
**Figure supplement 2.** Food-port entry and press rates during Pavlovian conditioning and PIT test for lOFC→BLA optical inhibition experiment.

To arbitrate between these possibilities, we multiplexed optogenetic and chemogenetic inhibition to perform a serial circuit disconnection. We disconnected lOFC→BLA projection activity during stimulus-outcome learning from BLA→lOFC projection activity during the retrieval of these memories at the PIT test (*Figure 5a*). For the disconnection group (*N* = 10), we again expressed ArchT bilaterally in lOFC neurons (*Figure 5b–d*) to allow expression in lOFC axons and terminals in the BLA. This time, we implanted the optical fiber only unilaterally in the BLA (*Figure 5b–d*), so that

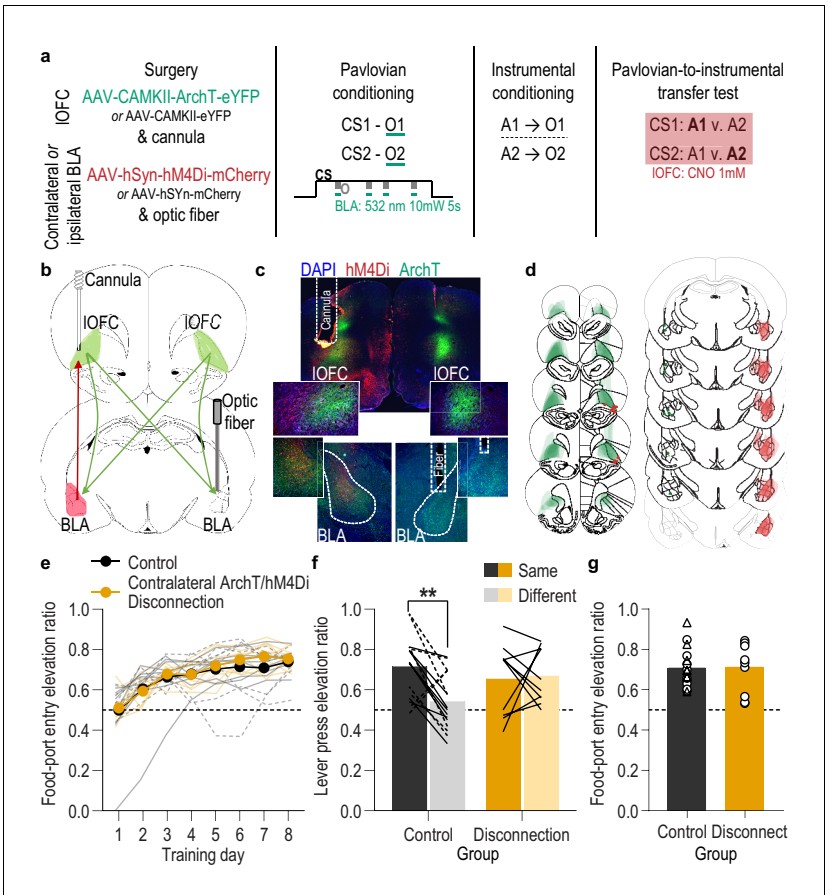

**Figure 5.** Serial disconnection of lOFC→BLA projections during stimulus-outcome pairing from BLA→lOFC projections during Pavlovian-to-instrumental transfer test disrupts stimulus-outcome memory. (a) Procedure schematic. CS, conditional stimulus (white noise or tone); O, outcome (sucrose solution or food pellet); A, action (left or right lever press); CNO, clozapine-n-oxide. (b) Schematic of multiplexed optogenetic/chemogenetic inhibition strategy for unilateral optical inhibition of lOFC→BLA projections during Pavlovian conditioning and contralateral, unilateral, chemogenetic inhibition of BLA→lOFC projections during the PIT test. (c) Top: Representative fluorescent image of bilateral ArchT-eYFP expression in lOFC cells bodies and unilateral expression of hM4Di-mCherry in BLA axons and terminals in the lOFC in the vicinity of implanted guide cannula. Bottom: Representative image of unilateral BLA fiber placement in the vicinity of immunofluorescent ArchT-eYFP expressing lOFC axons and terminals (right) and unilateral expression of hM4Di-mCherry in BLA cell bodies in the contralateral hemisphere (left). (d) Schematic representation of bilateral ArchT-eYFP expression and unilateral cannula placement in lOFC and unilateral hM4Di expression and placement of optical fiber tips in the contralateral BLA for all Contralateral group subjects. Fibers are shown in left and cannula placement in the right hemisphere, but fiber/cannula hemisphere arrangement was counterbalanced across subjects. See *Figure 5—figure supplement 1* for histological verification of ipsilateral control. (e) Elevation [(CS probe entry rate)/(CS probe entry rate + preCS entry rate)] in food-port entries during the CS probe period (after CS onset, before first reward delivery), averaged across trials and CSs for each day of Pavlovian conditioning. Thin light lines represent individual subjects (Contralateral eYFP/mCherry (solid lines) and Ipsilateral ArchT/hM4Di (dashed lines) collapsed into a single control group. (f) Elevation in lever presses on the lever that earned the same outcome as the presented CS (Same; [(presses on Same lever during CS)/(presses on Same lever during CS + Same presses during preCS)], averaged across trials and across CSs), relative to the elevation in responding on the alternate lever (Different; [(presses on Different lever during CS)/(presses on Different lever during CS + Different presses during preCS)], averaged across trials and across CSs) during the PIT test. Lines represent individual subjects. (g) Elevation in food-port entries to CS presentation (averaged across trials and CSs) during the PIT test. Data points represent individual subjects, triangles indicate ipsilateral control subjects. Control, *N* = 16; Contralateral disconnection group, *N* = 10. **p<0.01. See *Figure 5—source data 1*.

The online version of this article includes the following source data and figure supplement(s) for figure 5:

**Source data 1.** Source data for *Figure 5* and *Figure 5—figure supplement 1–2*.

*Figure 5 continued on next page*

*Figure 5 continued*

**Figure supplement 1.** Histological verification for unilateral, ipsilateral lOFC→BLA/BLA→lOFC inhibition subjects.
**Figure supplement 2.** Food-port entry and press rates during Pavlovian conditioning and PIT test for lOFC→BLA/BLA→lOFC serial disconnection experiment.

green light (532 nm, ~10 mW), delivered again during Pavlovian conditioning for 5 s during the delivery and retrieval of each reward during each cue, would inhibit both the ipsilateral and contralateral lOFC input to the BLA of only one hemisphere. In these subjects, we also expressed the inhibitory designer receptor human M4 muscarinic receptor (hM4Di) unilaterally in the BLA of the hemisphere opposite to the optical fiber and in that same hemisphere placed a guide cannula over the lOFC near hM4Di-expressing BLA axons and terminals (*Figure 5b–d*). This allowed us to infuse the hM4Di ligand clozapine-*n*-oxide (CNO; 1 mM in 0.25 µl) prior to the PIT test to unilaterally inhibit BLA terminals in the lOFC, which are largely ipsilateral (*Lichtenberg et al., 2017*), in the hemisphere opposite to that for which we had inhibited lOFC→BLA projection activity during Pavlovian conditioning. Thus, we optically inhibited the lOFC→BLA stimulus-outcome learning pathway in one hemisphere at each stimulus-outcome pairing during Pavlovian conditioning, and chemogenetically inhibited the putative BLA→lOFC retrieval pathway in the opposite hemisphere during the PIT test in which stimulus-outcome memories must be used to guide choice. If BLA→lOFC projection activity mediates the retrieval of the sensory-specific associative memory that requires activation of lOFC→BLA projections to be encoded then we will have bilaterally disconnected the circuit, attenuating encoding in one hemisphere and retrieval in the other, thereby disrupting the ability to use the stimulus-outcome memories to guide choice behavior during the PIT test. If, however, these pathways mediate parallel information streams, that is, independent components of the stimulus-outcome memory, the expression of PIT should be intact because one of each pathway is undisrupted to mediate its individual component during each phase. The control group received identical procedures with the exception that viruses lacked ArchT and hM4Di (*N* = 8). To control for unilateral inhibition of each pathway without disconnecting the circuit, a second control group (*N* = 8) received the same procedures as the experimental contralateral ArchT/hM4Di disconnection group, except with BLA hM4Di and the lOFC guide cannula in the same hemisphere as the optical fiber used to inactivate lOFC axons and terminals in the BLA (*Figure 5—figure supplement 1*). Thus, during the PIT test for this group the BLA→lOFC pathway was chemogenetically inactivated in the same hemisphere in which the lOFC→BLA pathway had been optically inactivated during Pavlovian conditioning, leaving the entire circuit undisrupted in the other hemisphere. These control groups did not differ on any measure and so were collapsed into a single control group [(Pavlovian training, Training: $F_{(2.2,31.3)}$ = 12.96, p<0.0001; Control group type: $F_{(1,14)}$ = 0.02, p=0.89; Group x Training: $F_{(7.98)}$ = 0.76, p=0.62) (PIT Lever presses, Lever: $F_{(1,14)}$ = 14.68, p=0.002; Control group type: $F_{(1,14)}$ = 0.38, p=0.55; Group x Lever: $F_{(1,14)}$ = 0.43, p=0.52) (PIT Food-port entries, $t_{14}$ = 0.72, p=0.48)]. See also *Figure 5—figure supplement 2* for disaggregated control data.

We found evidence that activity in lOFC→BLA projections mediates the encoding of the sensory-specific stimulus-outcome memory that is later used to allow cues to guide choice via activation of BLA→lOFC projections. As with the bilateral inhibition experiments, the control and disconnection groups both developed a Pavlovian conditional goal-approach response with training (*Figure 5e*; Training: $F_{(2.8,68.1)}$ = 28.13, p<0.0001; Group (Combined control group v. Contralateral ArchT/hM4Di- disconnection): $F_{(1,24)}$ = 0.46, p=0.51; Group x Training: $F_{(7,168)}$ = 0.44, p=0.88; see also *Figure 5—figure supplement 2a*), which was similarly expressed during the PIT test (*Figure 5g*; $t_{24}$ = 0.11, p=0.91; see also *Figure 5—figure supplement 2d*). But disconnection of lOFC→BLA projection activity during stimulus-outcome learning from BLA→lOFC projection activity during the PIT test attenuated the ability to use such memories to guide choice behavior (*Figure 5f*; Group x Lever: $F_{(1,24)}$ = 5.57, p=0.03; Group: $F_{(1,24)}$ = 0.47, p=0.50; Lever: $F_{(1,24)}$ = 1.39, p=0.21; see also *Figure 5—figure supplement 2c*). Whereas in the control group cue presentation significantly biased choice toward the action earning the same predicted reward, this outcome-specific PIT effect did not occur in the disconnection group. Rather, during the cues rats in the disconnection group showed a non-discriminate elevation in pressing on both levers (*Figure 5—figure supplement 2c*). Thus, disconnection of lOFC→BLA projection activity during stimulus-outcome learning from BLA→lOFC projection activity during the retrieval of this information attenuated the ability to use stimulus-outcome

memories to bias choice behavior, indicating that the lOFC and BLA form a bidirectional circuit for the encoding (lOFC→BLA) and use (BLA→lOFC) of appetitive stimulus-outcome memories.

## Discussion

Using fiber photometry bulk calcium imaging, cell-type and pathway-specific optogenetic inhibition, multiplexed optogenetic and chemogenetic inhibition, Pavlovian conditioning, and the outcome selective PIT test, we explored the function of the BLA and its interaction with the lOFC in the ability to learn detailed cue-reward memories and use them to guide decision making. Such memories are critical to the ability to use environmental cues to infer which specific rewards are likely to be available in the current state and, thus, to choose adaptively. We found that the BLA is robustly activated at the time of stimulus-outcome pairing and that this activity is necessary for sensory-specific, appetitive associative memories to be encoded, so that they can later influence decision making. We also found that this BLA activity is not necessary for the appetitive learning that supports general conditional goal-approach behavior, which does not require a detailed stimulus-outcome memory. lOFC input to the BLA supports its function in encoding stimulus-outcome memories and BLA projections back to the lOFC mediate the use of these memories to guide decision making. Thus, the lOFC→BLA→lOFC circuit regulates the encoding and subsequent use of the state-dependent and sensory-specific reward memories that are critical for decision making between two appetitive choices.

BLA neurons were found to be robustly activated at the time of stimulus-reward pairing as well as at cue onset, consistent with prior evidence that the BLA is activated by both rewards (*Crouse et al., 2020*; *Fontanini et al., 2009*; *Malvaez et al., 2019*; *Roesch et al., 2010*; *Schoenbaum et al., 1998*; *Sugase-Miyamoto and Richmond, 2005*) and their predictors (*Belova et al., 2008*; *Beyeler et al., 2018*; *Beyeler et al., 2016*; *Crouse et al., 2020*; *Lutas et al., 2019*; *Malvaez et al., 2015*; *Muramoto et al., 1993*; *Paton et al., 2006*; *Schoenbaum et al., 1998*; *Schoenbaum et al., 1999*; *Sugase-Miyamoto and Richmond, 2005*; *Tye and Janak, 2007*; *Tye et al., 2008*). Interestingly, the cues triggered a transient elevation in BLA activity at their onset, rather than a sustained elevation throughout their 2 min duration, perhaps suggesting that such activity reflects the state change, rather than the state per se. Both the cue and reward responses were present from the first conditioning session and persisted throughout training. That we detected cue responses on the first day of training before associative learning had occurred is, perhaps, unexpected and likely due to the novelty of the auditory stimuli during early training (*Bordi and LeDoux, 1992*; *Bordi et al., 1993*; *Cromwell et al., 2005*; *Romanski et al., 1993*). Indeed, we found that presentation of identical auditory stimuli unpaired with reward activated BLA neurons during the first session, much like the reward-predictive cues, but, in contrast to the reward-predictive cues, this response habituated over subsequent sessions. Thus, BLA cue responses later in training result from appetitive associative learning. Whereas we detected reward responses throughout training, prior data have demonstrated a shift in BLA responses from the reward to predictive events (*Crouse et al., 2020*) and little response to rewards in the absence of learning (*Malvaez et al., 2015*). The persistent reward response detected here likely results from the uncertainty of reward timing during the cues, which set the context for the intermittent availability of one specific reward. Another possibility is that it relates to the learning of two unique cue-reward contingencies, which was not the case in prior tasks. Nonetheless, the data show the BLA to be robustly activated at the time of stimulus-reward pairing in a task known to engender the encoding of detailed, sensory-specific stimulus-outcome memories.

We also found the BLA to be necessary, specifically at the time of stimulus-reward pairing, to encode the detailed stimulus-outcome memories. This is consistent with evidence that either pre- or post-training BLA lesion or pre-test inactivation disrupts appetitive conditional behaviors that rely on a sensory-specific, stimulus-outcome memory in rodents (*Blundell et al., 2001*; *Corbit and Balleine, 2005*; *Derman et al., 2020*; *Hatfield et al., 1996*; *Lichtenberg et al., 2017*; *Lichtenberg and Wassum, 2017*; *Malvaez et al., 2015*; *Morse et al., 2020*; *Ostlund and Balleine, 2008*) and in primates (*Murray and Izquierdo, 2007*; *Málková et al., 1997*). Leveraging the temporal resolution of optogenetics, we demonstrated that BLA principal neurons mediate the *encoding* of such memories, and specifically that activity at the time of reward experience during a cue is critical. Inhibiting the BLA during reward experience attenuated the animal's ability to link that specific rewarding event to the associated cue, disrupting the encoding of the sensory-specific stimulus-outcome memories to the extent that animals were unable to later use those memories to guide choice behavior. Future work

is needed to reveal the precise information content encoded by BLA neurons during reward experience that confers their necessary function in the formation of stimulus-outcome memories, although BLA neurons will respond selectively to unique food rewards (*Liu et al., 2018*), which could support the generation of sensory-specific reward memories. Whether BLA cue responses are also important for encoding stimulus-outcome memories is another important question exposed by the current results.

BLA activity during stimulus-outcome pairing was critical for encoding a detailed, outcome-specific, appetitive cue-reward memory, but it was not necessary for the learning underlying the development a non-specific Pavlovian conditional goal-approach response, consistent with data collected with BLA lesions or inactivation (*Corbit and Balleine, 2005*; *Everitt et al., 2000*; *Hatfield et al., 1996*; *Malvaez et al., 2015*; *Morse et al., 2020*; *Parkinson et al., 2000*). Although influenced by positive outcome valence, such responses do not require a rich sensory-specific representation of the predicted reward. Thus, BLA neurons appear not to be required to reinforce an appetitive Pavlovian response policy. Rather, the BLA mediates the encoding of the association between a cue and the specific reward it predicts, which includes encoding of the sensory-specific features of the reward. Optical stimulation of BLA neurons will, however, augment conditional goal-approach responses (*Servonnet et al., 2020*), suggesting BLA activation is capable of influencing such appetitive conditional behaviors.

Input from the lOFC was found to facilitate the BLA's function in mediating the encoding of stimulus-outcome memories. This expands upon previous findings that pre-training lOFC lesions disrupt behaviors that require a sensory-specific stimulus-outcome memory (*Izquierdo et al., 2004*; *Machado and Bachevalier, 2007*; *Ostlund and Balleine, 2007a*; *Pickens et al., 2005*; *Pickens et al., 2003*; *Rhodes and Murray, 2013*; *Scarlet et al., 2012*), that the lOFC is active during cue-reward learning (*Constantinople et al., 2019*; *Miller et al., 2018*; *Paton et al., 2006*; *Schoenbaum et al., 1998*; *Takahashi et al., 2013*; *Wallis and Miller, 2003*), and that encoding of expected outcomes in the BLA requires an intact lOFC (*Lucantonio et al., 2015*; *Saddoris et al., 2005*). Our data add to this literature by revealing the causal contribution of the direct lOFC→BLA pathway, specifically at the time of stimulus-outcome pairing, to the formation of detailed, outcome-specific, appetitive associative memories. Indeed, lOFC neurons respond to rewarding events during learning to signal reward expectations that may support learning in downstream structures, such as the BLA (*Stalnaker et al., 2007*; *Stalnaker et al., 2018*). Prior evidence also indicates that activity in lOFC→BLA projections drives the encoding of the incentive value of a specific rewarding event (*Malvaez et al., 2019*). Such incentive value is dependent upon one's current physiological state (e.g. food has high value when hungry, but low when sated). Thus, lOFC→BLA projections may be responsible for linking states, defined by internal physiological and external predictive cues, to the specific rewarding events with which they are associated. The precise information content conveyed by lOFC→BLA projections and how it is used in the BLA is a critical question for follow-up investigation.

We also discovered that the lOFC and BLA form a bidirectional circuit for the encoding and use of appetitive stimulus-outcome memories. The BLA has been implicated in appetitive decision making (*Costa et al., 2016*; *Costa et al., 2019*; *Izquierdo et al., 2013*; *Johnson et al., 2009*; *Orsini et al., 2017*; *Ostlund and Balleine, 2008*; *Stolyarova et al., 2019*; *Wellman et al., 2005*) and has been shown in non-human primates to interact with the lOFC in that regard (*Baxter et al., 2000*; *Fiuzat et al., 2017*). We previously found that BLA activity correlates with and regulates the ability to use sensory-specific, appetitive, stimulus-outcome memories to guide choice behavior (*Malvaez et al., 2015*). This function is mediated via direct BLA→lOFC projections, but does not require activation of lOFC→BLA projections (*Lichtenberg et al., 2017*). Here, using a serial disconnection procedure, we found that during reward choice BLA→lOFC projection activity mediates the use of the sensory-specific associative information that is learned via activation of lOFC→BLA projections. Thus, lOFC→BLA→lOFC is a functional circuit for the encoding (lOFC→BLA) and subsequent use (BLA→lOFC) of sensory-specific reward memories to inform decision making. Interestingly, the serial disconnection disrupted the outcome-specificity of PIT but, unlike bilateral BLA or lOFC→BLA inhibition during learning, allowed the cues to non-discriminately excite instrumental activity. This could have resulted from incomplete disconnection. But it may indicate that lOFC→BLA projections facilitate the encoding of a broader set of information than that being transmitted back to the lOFC by BLA→lOFC projection activity during choice. BLA→lOFC projections mediate use of the sensory-

specific components of the reward memory needed to allow animals to know during a cue which specific reward is predicted and thus which action to select, but lOFC→BLA projections may facilitate the encoding of additional features of the memory, including those capable of promoting food- or reward-seeking activity more broadly. The encoding of such information would have been disrupted by bilateral lOFC→BLA or BLA inactivation during learning, but in the disconnection experiment could have been learned in the hemisphere that did not receive lOFC→BLA inactivation and subsequently retrieved via an alternate BLA pathway. Indeed, BLA→lOFC are not the only amygdala projections involved in reward memory (*Beyeler et al., 2016*; *Corbit et al., 2013*; *Fisher et al., 2020*; *Kochli et al., 2020*; *Morse et al., 2020*; *Parkes and Balleine, 2013*). lOFC activity in both humans and non-human animals can encode the features of an expected reward (*Howard et al., 2015*; *Howard and Kahnt, 2018*; *Klein-Flügge et al., 2013*; *Lopatina et al., 2015*; *McDannald et al., 2014*; *Pritchard et al., 2005*; *Suzuki et al., 2017*; *van Duuren et al., 2007*; *Zhou et al., 2019*) and the lOFC has been proposed to be critical for using this information to guide decision making (*Bradfield and Hart, 2020*; *Delamater, 2007*; *Groman et al., 2019*; *Keiflin et al., 2013*; *Rich and Wallis, 2016*; *Rudebeck and Rich, 2018*; *Rudebeck and Murray, 2014*; *Sharpe and Schoenbaum, 2016*; *Wilson et al., 2014*). This might especially be the case in novel situations (*Gardner and Schoenbaum, 2020*). The PIT test is a novel choice scenario in which the subjects must use the cues to represent the sensory-specific features of the predicted reward, infer which reward is most likely to be available and, therefore, which action will be the most beneficial. lOFC→-BLA projection activity, perhaps via relaying reward expectation (*Stalnaker et al., 2007*; *Stalnaker et al., 2018*), regulates the associative learning that may allow subsequent activity in BLA→lOFC projections to promote the representation of a specific predicted reward in the lOFC to enable decision making. The precise information content conveyed by each component of the lOFC-BLA circuit and how it is used in the receiving structure is a critical follow-up question that will require a cellular resolution investigation of the activity of each pathway. Another critical question is whether this circuitry similarly mediates appetitive associative learning and its influence on decision making in females. The exclusion of female subjects is a clear limitation of this study, though females do show similar performance in the task used here and also require the BLA and lOFC for its performance (*Ostlund and Balleine, 2007a*; *Ostlund and Balleine, 2008*). Whether this lOFC-BLA architecture also underlies sensory-specific aversive memory is also a question ripe for further exploration.

The BLA, via input from the lOFC, helps to link environmental cues to the specific rewards they predict and, via projections back to the lOFC, to allow the cues to access those representations to influence decision making. An inability to either properly encode reward memories or to use such memories to inform decision making can lead to ill-informed motivations and decisions. This is characteristic of the cognitive symptoms underlying many psychiatric diseases, including substance use disorder. The OFC-BLA circuit is known to be altered by addictive substances (*Arguello et al., 2017*) and to be dysfunctional in myriad psychiatric illnesses (*Goldstein and Volkow, 2011*; *Liu et al., 2014*; *Passamonti et al., 2012*; *Ressler and Mayberg, 2007*; *Sladky et al., 2015*). Thus, these data may also aid our understanding and treatment of substance use disorder and other mental illnesses marked by disruptions to decision making.

## Materials and methods

### Key resources table

| Reagent type (species) or resource | Designation | Source or reference | Identifiers | Additional information |
|---|---|---|---|---|
| Recombinant DNA reagent | pENN.AAV5.CAMKII. GCaMP6f.WPRE.SV40 | Addgene | Cat: 100834-AAV5 RRID:Addgene_100834 | Lot # v59618 |
| Recombinant DNA reagent | rAAV5-CAMKIIa-eArchT3.0-eYFP | UNC-CH vector core | Deisseroth | Lot # V4883D |
| Recombinant DNA reagent | rAAV5-CAMKIIa-eYFP | UNC-CH vector core | Deisseroth | Lot # AV4808I |
| Recombinant DNA reagent | pAAV8-hSyn-hM4D (Gi)-mCherry | Addgene | Cat: 50475-AAV8 RRID:Addgene_50475 | Lot # v5483 |

*Continued on next page*

*Continued*

| Reagent type (species) or resource | Designation | Source or reference | Identifiers | Additional information |
|---|---|---|---|---|
| Recombinant DNA reagent | pAAV8-hSyn-mCherry | Addgene | Cat: 114472-AAV8 RRID:Addgene_114472 | |
| Other | Optical fiber (photometry) | Neurophotometrics | | Diameter: 200 µm; NA: 0.37; Length: 8–8.5 mm |
| Other | Optical fiber (manipulation) | Thorlabs | Cat: FT200UMT | Core: 200 µm; NA: 0.39; Length: 8–8.5 mm |
| Other | Optical ferrules | Kientec | Cat: FAZI-LC-230 | |
| Other | Guide cannula | Plastics One | Cat: C313G/SPC | Length: cut to 4 mm below pedestal |
| Chemical compound, drug | Clozapine *N-oxide* | Tocris | Cat: 4936/10 CAS: 34233-69-7 | |
| Other | Dustless precision Chocolate-flavored purified pellets | Bio-Serv | Cat: F0299 | 45 mg |
| Other | Sucrose | Ralphs | UPC: 0001111083805 | |
| Antibody | Chicken polyclonal anti-GFP antibody | Abcam | Cat: ab13970 | (1:1000) |
| Antibody | Goat polyclonal anti-chicken IgG, Alexa Fluor 488 conjugate | Abcam | Cat: ab150169 RRID:AB_2636803 | (1:500) |
| Antibody | Rabbit polyclonal anti-DsRed antibody | Takara Bio | Cat: 632496 RRID:AB_10013483 | (1:1000) |
| Antibody | Goat polyclonal anti-rabbit IgG, Alexa Fluor 594 conjugate | Invitrogen | Cat: A-11012 RRID:AB_253407 | (1:500) |
| Other | ProLong Gold Antifade Mountant with DAPI | Invitrogen | Cat: P36931 | |
| Chemical compound, drug | Paraformaldehyde | Sigma | Cat: P6148 | |
| Software, algorithm | MED-PC IV | Med Associates, Inc | RRID:SCR_012156 | |
| Software, algorithm | GraphPad Prism | GraphPad Software | RRID:SCR_002798 | Version: 8 |
| Software, algorithm | MatLab | MathWorks | RRID:SCR_001622 | Version: 2019a |
| Software, algorithm | SPSS | IBM | RRID:SCR_019096 | Version: 26 |
| Software, algorithm | Bonsai | Bonsai | RRID:SCR_017218 | Version: 2.3 |
| Software, algorithm | Minianalysis | Synaptosoft | RRID:SCR_002184 | Version 6 |
| Software, algorithm | BZ-X Analyze software | Keyence | RRID:SCR_017205 | |
| Software, algorithm | Zeiss Zen Blue software | Zeiss | RRID:SCR_013672 | |
| Software, algorithm | Illustrator | Adobe | RRID:SCR_010279 | |
| Software, algorithm | ImageJ | NIH | RRID:SCR_003070 | |
| Software, algorithm | Excel | Microsoft | RRID:SCR_016137 | |

## Subjects

Male, Long Evans rats aged 8–10 weeks at the start of the experiment (Charles River Laboratories, Wilmington, MA) were group housed (2/cage) in a temperature (68–79 °F) and humidity (30–70%) regulated vivarium prior to surgery and then subsequently housed individually to preserve implants. Rats were provided with water ad libitum in the home cage and were maintained on a food-restricted 12–14 g daily diet (Lab Diet, St. Louis, MO) to maintain ~85–90% free-feeding body weight. Rats were handled for 3–5 days prior to the onset of each experiment. Separate groups of naive rats were used for each experiment. Experiments were performed during the dark phase of a 12:12 hr reverse dark/light cycle (lights off at 7AM). All procedures were conducted in accordance with the NIH Guide for the Care and Use of Laboratory Animals and were approved by the UCLA Institutional Animal Care and Use Committee.

## Surgery

Standard surgical procedures, described previously (*Lichtenberg et al., 2017*; *Malvaez et al., 2015*; *Malvaez et al., 2019*), were used for all surgeries. Rats were anesthetized with isoflurane (4–5% induction, 1–3% maintenance) and a nonsteroidal anti-inflammatory agent was administered pre- and post-operatively to minimize pain and discomfort.

### Fiber photometry recordings

Surgery occurred prior to onset of behavioral training. Rats ($N$ = 11) were infused bilaterally with adeno-associated virus (AAV) expressing the genetically encoded calcium indicator GCaMP6f under control of the calcium/calmodulin-dependent protein kinase (CaMKII) promoter (pENN.AAV5.CAM-KII.GCaMP6f.WPRE.SV40, Addgene, Watertown, MA) to drive expression preferentially in principal neurons. Virus (0.5 µl) was infused a rate of 0.1 µl/min into the BLA [AP: −2.7 ($N$ = 5) or −3.0 ($N$ = 6); ML:±5.0; DV: −8.6 mm from bregma] using a 28-gauge injector. Injectors were left in place for an additional 10 min to ensure adequate diffusion and to minimize off-target spread along the injector tract. Optical fibers (200 µm diameter, 0.37 numerical aperture (NA), Neurophotometrics, San Diego, CA) were implanted bilaterally 0.2 mm dorsal to the infusion site to allow subsequent imaging of GCaMP fluctuations in BLA neurons. These procedures were replicated in a separate group of subjects ($N$ = 6) that served as unpaired CS$_\varnothing$ control. Behavioral training commenced approximately 3–4 weeks after surgery to allow sufficient expression in BLA neurons.

### Optogenetic inhibition of BLA

Prior to the onset of behavioral training, rats were randomly assigned to a viral group and were infused bilaterally with AAV encoding either the inhibitory opsin archaerhodopsin T (ArchT; $N$ = 9; rAAV5-CAMKIIa-eArchT3.0-eYFP, University of North Carolina Vector Core, Chapel Hill, NC) or the enhanced yellow fluorescent protein control (eYFP; $N$ = 10; rAAV5-CAMKIIa-eYFP, University of North Carolina Vector Core) under control of the CaMKII promoter. Virus (0.5 µl) was infused at a rate of 0.1 µl/min into the BLA (AP: −2.8; ML:±5.0; DV: −8.6 mm from bregma) using a 28-gauge injector. Injectors were left in place for an additional 10 min. Optical fibers (200 µm core, 0.39 NA, Thorlabs, Newton, NJ) held in ceramic ferrules (Kientec Systems, Stuart, FL) were implanted bilaterally 0.6 mm dorsal to the injection site to allow subsequent light delivery to ArchT- or eYFP-expressing BLA neurons. Identical surgical procedures were used for a separate yoked inhibition control group ($N$ = 7). A third group ($N$ = 5) also received bilateral infusion of rAAV5-CAMKIIa-eArchT3.0-eYFP into the BLA, without fiber implants, for subsequent ex vivo electrophysiological validation of optical inhibition of BLA neurons. Experiments commenced 3 weeks after surgery to allow sufficient expression in BLA neurons.

### Optogenetic inhibition of lOFC→BLA projections

Prior to the onset of behavioral training, rats were randomly assigned to a viral group and were infused with AAV encoding either the inhibitory opsin ArchT ($N$ = 8; rAAV5-CAMKIIa-eArchT3.0-eYFP) or eYFP control ($N$ = 8; rAAV5-CAMKIIa-eYFP). Virus (0.3 µl) was infused at a rate of 0.1 µl/min bilaterally into the lOFC (AP:+3.3; ML:±2.5; DV: −5.4 mm from bregma) using a 28-gauge injector tip. Injectors were left in place for an additional 10 min. Optical fibers (200 µm core, 0.39 NA) held in ceramic ferrules were implanted bilaterally in the BLA (AP: −2.7; ML:±5.0; DV: −8.0 mm from bregma) to allow subsequent light delivery to ArchT- or eYFP-expressing axons and terminals in the BLA. A separate group ($N$ = 4) also received bilateral infusion of rAAV5-CAMKIIa-eArchT3.0-eYFP into the lOFC, without fiber implants, for subsequent ex vivo electrophysiological validation of optical inhibition of lOFC terminals in the BLA. Experiments began 7–8 weeks following surgery to allow axonal transport to the BLA.

### Multiplexed optogenetic inhibition lOFC→BLA projections and chemogenetic inhibition of BLA→lOFC projections for serial circuit disconnection

Prior to the onset of behavioral training, rats were randomly assigned to viral group. The disconnection group ($N$ = 10) was infused with AAV encoding the inhibitory opsin ArchT (rAAV5-CAMKIIa-

eArchT3.0-eYFP; 0.3 µl) bilaterally at a rate of 0.1 µl/min into the lOFC (AP:+3.3; ML:±2.5; DV: −5.4 mm from bregma) using a 28-gauge injector tip. Injectors were left in place for an additional 10 min. An optical fiber (200 µm core, 0.39 NA) held in a ceramic ferrule was implanted unilaterally (hemisphere counterbalanced across subjects) in the BLA (AP: −2.7; ML:±5.0; DV: −7.7 mm from dura) to allow subsequent light delivery to both the ipsilateral and contralateral ArchT-expressing axons and terminals in the BLA of only one hemisphere. During the same surgery, in the hemisphere contralateral to optical fiber placement, a second AAV was infused unilaterally at a rate of 0.1 µl/min into the BLA (AP: −3.0; ML:±5.1; DV: −8.6 from bregma) to drive expression of the inhibitory designer receptor *human M4 muscarinic receptor* (hM4Di; pAAV8-hSyn-hM4D(Gi)-mCherry, Addgene; 0.5 µl). A 22-gauge stainless-steel guide cannula was implanted unilaterally above the lOFC (AP:+3.0; ML:±3.2: DV: −4.0) of the BLA-hM4Di hemisphere to target the hM4D(Gi)-expressing axonal terminals, which are predominantly ipsilateral. This allowed subsequent optical inhibition of lOFC terminals in the BLA of one hemisphere and chemogenetic inhibition of BLA terminals in the lOFC of the other hemisphere, thus disconnecting the putative lOFC→BLA→lOFC circuit. Surgical procedures were identical for the fluorophore-only control group (N = 8), except with AAVs encoding only eYFP (lOFC; rAAV5-CAMKIIa-eYFP) and mCherry (BLA; pAAV8-hSyn-mCherry). A separate ipsilateral control group received the same surgical procedures as the experimental contralateral ArchT/hM4Di group, but with BLA pAAV8-hSyn-hM4D(Gi)-mCherry and lOFC guide cannula placed in the same hemisphere as the BLA optical fiber. Experiments began 7–8 weeks following surgery to allow sufficient viral expression and axonal transport. Two subjects became ill before testing and, thus, were excluded from the experiment (Contralateral ArchT/hM4Di, N = 1; Ipsilateral ArchT/hM4Di, N = 1).

## Behavioral procedures

### Apparatus

Training took place in Med Associates conditioning chambers (East Fairfield, VT) housed within sound- and light-attenuating boxes, described previously (*Collins et al., 2019*; *Malvaez et al., 2015*; *Malvaez et al., 2019*). For optogenetic manipulations, the chambers were outfitted with an Intensity Division Fiberoptic Rotary Joint (Doric Lenses, Quebec, QC, Canada) connecting the output fiber optic patch cords to a laser (Dragon Lasers, ChangChun, JiLin, China) positioned outside of the chamber.

Each chamber contained two retractable levers that could be inserted to the left and right of a recessed food-delivery port (magazine) in the front wall. A photobeam entry detector was positioned at the entry to the food port. Each chamber was equipped with a syringe pump to deliver 20% sucrose solution in 0.1 ml increments through a stainless-steel tube into one well of the food port and a pellet dispenser to deliver 45 mg purified chocolate food pellets (Bio-Serv, Frenchtown, NJ) into another well. Both a tone and white noise generator were attached to individual speakers on the wall opposite the levers and food-delivery port. A 3-watt, 24-volt house light mounted on the top of the back wall opposite the food-delivery port provided illumination and a fan mounted to the outer chamber provided ventilation and external noise reduction. Behavioral procedures were similar to that we have described previously (*Lichtenberg et al., 2017*; *Lichtenberg and Wassum, 2017*; *Malvaez et al., 2015*).

### Magazine conditioning

Rats first received one day of training to learn where to receive the sucrose and food pellet rewards. This included two separate sessions, separated by approximately 1 hr, order counterbalanced, one with 30 non-contingent deliveries of sucrose (60 s intertrial interval, ITI) and one with 30 food pellet deliveries (60 s ITI).

### Pavlovian conditioning

Rats then received 8 sessions of Pavlovian conditioning (one session/day on consecutive days) to learn to associate each of two auditory conditional stimuli (CSs; 80–82 db, 2 min duration), tone (1.5 kHz) or white noise, with a specific food reward, sucrose (20%, 0.1 ml/delivery) or purified chocolate pellets (45 mg; Bio-Serv). CS-reward pairings were counterbalanced at the start of each experiment. For half the subjects, tone was paired with sucrose and noise with pellets, with the other half receiving the opposite arrangement. Each session consisted of eight tone and eight white noise

presentations, with the exception of the fiber photometry experiments, in which rats received four of each CS/session to reduce session time and, thus, minimize the effects of photobleaching. During each 2 min CS, the associated reward was delivered on a 30 s random-time schedule, resulting in an average of 4 stimulus-reward pairings per trial. For the fiber photometry experiments, there was a minimum 15 s probe period after CS onset before the first reward delivery to allow us to dissociate signal fluctuations due to CS onset from those due to reward delivery/retrieval. CSs were delivered pseudo-randomly with a variable 2–4 min ITI (mean = 3 min).

Procedures were identical for the unpaired $CS_\varnothing$ control fiber photometry experiment, except no rewards were delivered during Pavlovian training. Subjects in this experiment instead received rewards in their home cage several hours after the $CS_\varnothing$ sessions. On the day following the last $CS_\varnothing$ session, these subjects received one session with non-contingent, unpredicted deliveries of sucrose and food pellets, each delivered on a 30 s random-time schedule during 4, 2 min periods (variable 2–4 min ITI, mean = 3 min), resulting in an average of 16 deliveries of each outcome.

## Instrumental conditioning

Rats were then given 11 days, minimum, of instrumental conditioning. They received two separate training sessions per day, one with the left lever and one with the right lever, separated by at least 1 hr. Each action was reinforced with a different outcome (e.g. left press-chocolate pellets/right press-sucrose solution; counterbalanced with respect to the Pavlovian contingencies). Each session terminated after 30 outcomes had been earned or 45 min had elapsed. Actions were continuously reinforced on the first day and then escalated ultimately to a random-ratio 20 schedule of reinforcement.

## Outcome-selective Pavlovian-to-instrumental transfer test

Following Pavlovian and instrumental conditioning, rats received an outcome-selective Pavlovian-to-instrumental transfer (PIT) test. On the day prior to the PIT test, rats were given a single 30 min extinction session during which both levers were available but pressing was not reinforced to establish a low level of responding. During the PIT test, both levers were continuously present, but pressing was not reinforced. After 5 min of lever-pressing extinction, each 2 min CS was presented separately four times in pseudorandom order, separated by a fixed 4 min inter-trial interval. No rewards were delivered during CS presentation.

## Data collection

Lever presses and/or discrete entries into the food-delivery port were recorded continuously for each session. For both Pavlovian training and PIT test sessions, the 2 min periods prior to each CS onset served as the baseline for comparison of CS-induced elevations in lever pressing and/or food-port entries.

## In vivo fiber photometry

Fiber photometry was used to image bulk calcium activity in BLA neurons throughout each Pavlovian conditioning session. We simultaneously imaged GCaMP6f and control fluorescence in the BLA using a commercial fiber photometry system (Neurophotometrics Ltd., San Diego, CA). Two light-emitting LEDs (470 nm: $Ca^{2+}$-dependent GCaMP fluorescence; 415 nm: autofluorescence, motion artifact, $Ca^{2+}$-independent GCaMP fluorescence) were reflected off dichroic mirrors and coupled via a patch cord (fiber core diameter, 200 µm; Doric Lenses) to the implanted optical fiber. The intensity of the light for excitation was adjusted to ~80 µW at the tip of the patch cord. Fluorescence emission was passed through a 535 nm bandpass filter and focused onto the complementary metal-oxide semiconductor (CMOS) camera sensor through a tube lens. Samples were collected at 20 Hz, interleaved between the 415 and 470 excitation channels, using a custom Bonsai (*Lopes et al., 2015*) workflow. Time stamps of task events were collected simultaneously through an additional synchronized camera aimed at the Med Associates interface, which sent light pulses coincident with task events. Signals were saved using Bonsai software and exported to MATLAB (MathWorks, Natick, MA) for analysis. Recordings were collected unilaterally from the hemisphere with the strongest fluorescence signal in the 470 channel at the start of the experiment, which was kept consistent throughout the

remainder of the experiment. Animals were habituated to the optical tether during the magazine conditioning sessions, but no light was delivered.

## Optogenetic inhibition of BLA neurons

Optogenetic inhibition was used to attenuate the activity of ArchT-expressing BLA neurons at the time of stimulus-outcome pairing during each CS during each Pavlovian conditioning session. Animals were habituated to the optical tether (200 µm, 0.22 NA, Doric) during the magazine conditioning sessions, but no light was delivered. During each Pavlovian conditioning session, green light (532 nm; 10 mW) was delivered to the BLA via a laser (Dragon Lasers, ChangChun) connected through a ceramic mating sleeve (Thorlabs) to the ferrule implanted on the rat. Light was delivered continuously for 5 s concurrent with each reward delivery. If the reward was retrieved (first food-port entry after reward delivery) while the light was still being delivered (i.e. within 5 s of reward delivery), then the light delivery was extended to 5 s from the time of the retrieval. If the reward was retrieved after the laser had gone off, then the retrieval entry triggered an additional 5 s continuous illumination. To control for the overall amount of inhibition, a separate control group received green light during the 2 min preCS baseline periods with the same number, duration, and pattern as the experimental group. Light effects were estimated to be restricted to the BLA based on predicted irradiance values (https://web.stanford.edu/group/dlab/cgi-bin/graph/chart.php). Following Pavlovian conditioning, rats proceeded through instrumental conditioning and the PIT test, as above. Light was not delivered during these subsequent phases of the experiment.

## Optogenetic inhibition of lOFC→BLA projections

Optogenetic inhibition was used to attenuate the activity of ArchT-expressing lOFC→BLA terminals at the time of stimulus-outcome pairing during each CS during each Pavlovian conditioning session. Procedures were identical to those for BLA inhibition above. Green light (532 nm; 10 mW) was delivered to the BLA continuously for 5 s concurrent with each reward delivery and/or retrieval during Pavlovian conditioning.

## Multiplexed optogenetic inhibition of lOFC→BLA projections during Pavlovian conditioning and chemogenetic inhibition of BLA→lOFC projections during the Pavlovian-to-instrumental transfer test for serial circuit disconnection

We multiplexed optogenetic inhibition of lOFC→BLA projection activity during stimulus-outcome pairing during Pavlovian conditioning with chemogenetic inhibition of BLA→lOFC projection activity during the PIT test to perform a serial circuit disconnection and ask whether activity in lOFC→BLA projections mediates the encoding of the stimulus-outcome memory that is later retrieved via activation of BLA→lOFC projections (*Lichtenberg et al., 2017*). That is, whether lOFC→BLA→lOFC is a functional circuit for the encoding (lOFC→BLA) and subsequent use for guiding decision making (BLA→lOFC) of appetitive, sensory-specific, stimulus-outcome memories. To achieve the serial circuit disconnection, in the experimental group, we optically inactivated ipsilateral and contralateral lOFC input to the BLA of only one hemisphere during stimulus-outcome pairing during Pavlovian conditioning, and then chemogenetically inactivated predominantly ipsilateral (*Lichtenberg et al., 2017*) BLA axons and terminals in the lOFC of the other hemisphere during the PIT test. This leaves one of each pathway undisrupted to mediate the stimulus-outcome learning (lOFC→BLA) and retrieval (BLA→lOFC), but if lOFC→BLA→lOFC forms a functional stimulus-outcome memory circuit, then we will have disconnected the circuit in each hemisphere.

## Optogenetic inhibition of lOFC→BLA projections during Pavlovian conditioning

Optogenetic inhibition was used to attenuate the activity of ArchT-expressing lOFC→BLA terminals of one hemisphere at the time of stimulus-outcome pairing (reward delivery and retrieval) during each CS during each Pavlovian conditioning session. Procedures were identical to those described above, except that green light (532 nm; 10 mW) was delivered unilaterally to the BLA continuously for 5 s concurrent with each reward delivery and retrieval during Pavlovian conditioning.

## Chemogenetic inhibition of BLA→lOFC projections during the Pavlovian-to-instrumental transfer test

Chemogenetic inhibition was used to inactivate hM4Di-expressing BLA axons and terminals in the lOFC of one hemisphere during the PIT test. For the contralateral ArchT/hM4Di group, chemogenetic inhibition occurred in the hemisphere opposite to the one that received optical inhibition of lOFC→BLA projections during learning, thus achieving the disconnection. In a separate ipsilateral control group, the chemogenetic inhibition occurred on the same side as optical inhibition of lOFC→BLA projections during learning, leaving the entire circuit undisrupted in one hemisphere, while controlling for unilateral inhibition of each pathway. We selected chemogenetic inhibition so it could be multiplexed with optogenetic inhibition and to allow inhibition throughout the duration of the PIT test. CNO (Tocris Bioscience, Sterling Heights, MI) was dissolved in aCSF to 1 mM and 0.25 µL was intracranially infused over 1 min into the lOFC as previously described (*Lichtenberg et al., 2017*). Injectors were left in place for at least one additional min to allow for drug diffusion. The PIT test commenced within 5–10 min following infusion. CNO dose was selected based on evidence of both its behavioral effectiveness and ability to attenuate the activity of hM4Di-expressing BLA terminals in the lOFC (*Lichtenberg et al., 2017*). We have also demonstrated that this dose of CNO when infused into the lOFC has no effect on reward-related behavior in the absence of the hM4Di transgene (*Lichtenberg et al., 2017*).

## Ex vivo electrophysiology

Whole-cell patch clamp recordings were used to validate the efficacy of optical inhibition of BLA principal neuron activity and lOFC terminal activity in the BLA. Recordings were performed in brain slices from ~3- to 4-month-old rats 3–4 (BLA cell body inhibition) or 7–8 (lOFC→BLA inhibition) weeks following surgery. To prepare brain slices, rats were deeply anesthetized with isoflurane and perfused transcardially with an ice-cold, oxygenated NMDG-based slicing solution containing (in mM): 30 NaHCO3, 20 HEPES, 1.25 NaH2PO4, 102 NMDG, 40 glucose, 3 KCl, 0.5 CaCl2-2H2O, 10 MgSO4-H2O (pH adjusted to 7.3–7.35, osmolality 300–310 mOsm/L). Brains were extracted and immediately placed in ice-cold, oxygenated NMDG slicing solution. Coronal slices (350 µm) were cut using a vibrating microtome (VT1000S; Leica Microsystems, Germany), transferred to an incubating chamber containing oxygenated NMDG slicing solution warmed to 32–34°C, and allowed to recover for 15 min before being transferred to an artificial cerebral spinal fluid (aCSF) solution containing (in mM): 130 NaCl, 3 KCl, 1.25 $NaH_2PO_4$, 26 $NaHCO_3$, 2 $MgCl_2$, 2 $CaCl_2$, and 10 glucose) oxygenated with 95% $O_2$, 5% $CO_2$ (pH 7.2–7.4, osmolality 290–310 mOsm/L, 32–34°C). After 15 min, slices were moved to room temperature and allowed to recover for ~30 additional min prior to recording. All recordings were performed using an upright microscope (Olympus BX51WI, Center Valley, PA) equipped with differential interference contrast optics and fluorescence imaging (QIACAM fast 1394 monochromatic camera with Q-Capture Pro software, QImaging, Surrey, BC, Canada). (Patch pipettes (3–5 MΩ resistance) contained a Cesium methanesulfonate-based internal recording solution (in mM): 125 Cs-methanesulfonate, 4 NaCl, 1 MgCl2, 5 MgATP, 9 EGTA, 8 HEPES, 1 GTP-Tris, 10 phosphocreatine, and 0.1 leupeptin; pH 7.2 with CsOH, 270–280 (mOsm). Biocytin (0.2%, Sigma-Aldrich, St. Louis, MO) was included in the internal recording solution for subsequent postsynaptic cell visualization and identification. Recordings were obtained using a MultiClamp 700B Amplifier (Molecular Devices, Sunnyvale, CA) and the pCLAMP 10.3 acquisition software.

## Validation of BLA principal neuron optogenetic inhibition

Whole-cell patch clamp recordings in current-clamp mode were obtained from BLA principal neurons expressing ArchT-eYFP (*N* = 12 cells, five subjects). Visible eYFP-expressing cell bodies were identified in the BLA for recordings. After breaking through the membrane, recordings were obtained from cells while injecting suprathreshold depolarizing current (1 s). Current injection intensities that resulted in 8–15 action potentials were selected for recordings (100–800 pA). Electrode access resistances were maintained at <30 MΩ. Green light (535 nm, 1 s pulse, 0.25–1 mW; CoolLED Ltd, Andover, UK) was delivered through the epifluorescence illumination pathway using Chroma Technologies filter cubes to activate ArchT and inhibit BLA cell bodies. The number of action potentials recorded in ArchT-expressing cells injected with suprathreshold current were recorded both prior to and after green light illumination.

## Validation of lOFC terminal optogenetic inhibition in the BLA

Whole-cell patch-clamp recordings were collected in voltage-clamp mode. Visible eYFP-expressing axons and terminals were identified in the BLA and recordings were obtained from postsynaptic BLA neurons located only in highly fluorescent regions. After breaking through the membrane, recordings were obtained while holding the membrane potential at −70 mV. Electrode access resistances were maintained at <30 MΩ. Spontaneous excitatory postsynaptic currents (sEPSCs) were recorded in the presence of the GABA$_A$ receptor antagonist bicuculline (10 µM). Fifteen seconds of baseline recordings of sEPSCs were obtained prior to exposure to green light. Following baseline measurements, recordings of sEPSCs were obtained during continuous exposure to green light (535 nm, 0.5 mW) for 15 s. Spontaneous EPSC events were analyzed offline using the automatic detection protocol within the MiniAnalysis software (Synaptosoft, version 6.0), and then were checked manually blinded to light condition.

## Histology

Following the behavioral experiments, rats were deeply anesthetized with Nembutal and transcardially perfused with phosphate buffered saline (PBS) followed by 4% paraformaldehyde (PFA). Brains were removed and post-fixed in 4% PFA overnight, placed into 30% sucrose solution, then sectioned into 30–40 µm slices using a cryostat and stored in PBS or cryoprotectant.

eYFP fluorescence was used to confirm ArchT expression in lOFC and BLA cell bodies. mCherry expression was used to confirm hM4D(Gi) in BLA cell bodies. Immunofluorescence was used to confirm expression of ArchT-eYFP in lOFC axons and terminals in the BLA. Floating coronal sections were washed 3 times in 1x PBS for 30 min and then blocked for 1–1.5 hr at room temperature in a solution of 3% normal goat serum and 0.3% Triton X-100 dissolved in PBS. Sections were then washed three times in PBS for 15 min and incubated in blocking solution containing chicken anti-GFP polyclonal antibody (1:1000; Abcam, Cambridge, MA) with gentle agitation at 4°C for 18–22 hr. Sections were next rinsed three times in PBS for 30 min and incubated with goat anti-chicken IgY, Alexa Fluor 488 conjugate (1:500; Abcam) at room temperature for 2 hr. Sections were washed a final three times in PBS for 30 min. Immunofluorescence was also used to confirm expression of hM4Di-mCherry in BLA axons and terminals in the lOFC. The signal for axonal expression of hM4D(Gi)-mCherry in terminals in the lOFC was immunohistochemically amplified following procedures described previously (*Lichtenberg et al., 2017*). Briefly, floating coronal sections were rinsed in PBS and blocked for 1–2 hr at room temperature in a solution of 10% normal goat serum and 0.5% Triton X-100 dissolved in PBS and then incubated in blocking solution containing rabbit anti-DsRed polyclonal antibody (1:1000; Takara Bio, Mountain View, CA) with gentle agitation at 4°C for 18–22 hr. Sections were next rinsed in blocking solution and incubated with goat anti-rabbit IgG, Alexa Fluor 594 conjugate (1:500; Invitrogen, Waltham, MA) for 2 hr. Slices were mounted on slides and coverslipped with ProLong Gold mounting medium with DAPI. Images were acquired using a Keyence BZ-X710 microscope (Keyence, El Segundo, CA) with a 4x, 10x, and 20x objective (CFI Plan Apo), CCD camera, and BZ-X Analyze software or a Zeiss apotome confocal microscope (Zeiss, Oberkochen, Germany) and Zeiss Zen Blue software (Zeiss). Subjects with off-target viral, fiber, and/or cannula placements were removed from the dataset (Fiber photometry: $N = 2$; Fiber photometry CS$_\varnothing$ control $N = 0$; BLA ArchT: $N = 2$; BLA ArchT yoked control: $N = 1$; Contralateral disconnection, $N = 6$; Ipsilateral control $N = 7$).

## Data analysis

### Behavioral analysis

Behavioral data were processed with Microsoft Excel (Microsoft, Redmond, WA). Left and/or right lever presses and/or entries into the food-delivery port were collected continuously for each training and test session. Acquisition of the Pavlovian conditional food-port approach response was assessed by computing an elevation ratio of the rate of entries into the food-delivery port (entries/min) during the CS prior to reward delivery (CS-probe) relative to 2 min baseline periods immediately prior to CS onset [(CS probe entry rate)/(CS probe entry rate +preCS entry rate)]. Data were averaged across trials for each CS and then averaged across the two CSs. We also compared the rate of food-port entries between the CS probe and the preCS baseline periods (see *Figure 1—figure supplements 1a*, *Figure 3—figure supplements 2a*, *Figure 4—figure supplements 2a*, *Figure 5—figure*

*supplements 2a*). Press rates on the last day of instrumental training were averaged across levers and compared between groups to test for any differences in the acquisition of lever press responding during instrumental training. No significant group differences were detected in any of the experiments (see *Figure 1—figure supplements 1b*, *Figure 3—figure supplements 2b*, *Figure 4—figure supplements 2b*, *Figure 5—figure supplements 2b*). For the PIT test, lever pressing during the 2 min baseline periods immediately prior to the onset of each CS was compared with that during the 2 min CS periods. For both the baseline and CS periods, lever pressing was separated for presses on the lever that, during training, earned the same outcome as the presented cue (i.e. preCS-Same and CS-Same presses) versus those on the other available lever (i.e. preCS-Different and CS-Different presses). To evaluate the influence of CS presentation on lever pressing, we computed an elevation ratio for each lever [(CS-Same presses)/(CS-Same presses + preCS Same presses)] and [(CS-Different presses)/(CS-Different presses + preCS Different presses)]. In all cases, there were no significant differences in baseline presses between levers in the absence of the CSs (Lever: lowest p=0.33, $F_{1,14} = 1.02$), and no effect of group on baseline lever pressing (Group: lowest p=0.54, $F_{2,23} = 0.63$; Group x Lever lowest p=0.21, $F_{1,14} = 1.71$). To evaluate the influence of CS presentation on food-port entries, that is, the conditional goal-approach responses, we also computed an elevation ratio [(CS entries)/(CS entries + preCS entries)]. Data were averaged across trials for each CS and then averaged across the two CSs. We also compared the rate of pressing on each lever and, separately, food-port entries between the CS and preCS baseline periods (see *Figure 1—figure supplements 1c–d*, *Figure 3—figure supplements 2c–d*, *Figure 4—figure supplements 2c–d*, *Figure 5—figure supplements 2c–d*).

## Fiber photometry data analysis

Data were pre-processed using a custom-written pipeline in MATLAB (MathWorks, Natick, MA). Data from the 415 nm isosbestic control channel were used to correct for motion artifacts and photobleaching. Using least-squares linear regression, the 415 signal was fit to the 470 signal. Change in fluorescence (ΔF/F) at each time point was calculated by subtracting the fitted 415 signal from the 470 signal and normalizing to the fitted 415 data [(470-fitted 415)/(fitted 415)] (See *Figure 1—figure supplement 2*). The ΔF/F data were then Z-scored [(ΔF/F - mean ΔF/F)/std(ΔF/F)]. Using a custom MATLAB workflow, Z-scored traces were then aligned to CS onset, reward delivery, reward retrieval (first food-port entry after reward delivery), and food-port entries without reward present during the CS probe period (after CS before first reward delivery) during the CS for each trial. Peak magnitude and AUC were calculated on the Z-scored trace for each trial using 3 s pre-event baseline and 3 s post-event windows. Data were averaged across trials and then across CSs. Session data were excluded if no transient calcium fluctuations were detected on the 470 nm channel above the isosbestic channel or if poor linear fit was detected due to excessive motion artifact. To examine the progression in BLA activity across training, we compared data across conditioning sessions 1, 2, 3/4, 5/6, and 7/8. Thus, data from the mid and latter training sessions were averaged across bins of two training sessions. Subjects without reliable data from at least one session per bin were excluded (CS +$N$ = 5; CS$_\varnothing N$ = 1). We were able to obtain reliable imaging data from all of the eight training sessions from $N$ = 8 of the 11 total final subjects that received CS-reward pairing (see *Figure 1—figure supplement 3*).

## Ex vivo electrophysiology

The number of action potentials evoked by suprathreshold current injection was compared before and during exposure to green light to confirm the inhibitory effect of ArchT in BLA principal neurons. To assess the effect of ArchT activation in lOFC→BLA terminals, the frequency of sEPSCs was compared before and during green light exposure.

## Statistical analysis

Datasets were analyzed by two-tailed, paired and unpaired Student's *t* tests, one-, two-, or three-way repeated-measures analysis of variance (ANOVA), as appropriate (GraphPad Prism, GraphPad, San Diego, CA; SPSS, IBM, Chicago, IL). Post hoc tests were corrected for multiple comparisons using the Bonferroni method. All data were tested for normality prior to analysis with ANOVA and

the Greenhouse-Geisser correction was applied to mitigate the influence of unequal variance between conditions. Alpha levels were set at $p<0.05$.

## Rigor and reproducibility

Group sizes were estimated a priori based on prior work using male Long Evans rats in this behavioral task (*Lichtenberg et al., 2017*; *Lichtenberg and Wassum, 2017*; *Malvaez et al., 2015*) and to ensure counterbalancing of CS-reward and Lever-reward pairings. Investigators were not blinded to viral group because they were required to administer virus. All behaviors were scored using automated software (MedPC). Each primary experiment included at least one replication cohort and cohorts were balanced by viral group, CS-reward and Lever-reward pairings, hemisphere etc. prior to the start of the experiment.

## Acknowledgements

We thank Dr. Avishek Adhikari for assistance setting up fiber photometry. We acknowledge the very helpful feedback from Dr. Alicia Izquierdo, Dr. Melissa Sharpe, Dr. Melissa Malvaez, and Dr. Avishek Adhikari on this manuscript. Lastly, we acknowledge the generous infrastructure support from the Staglin Center for Behavior and Brain Sciences.

## Additional information

### Competing interests

Kate M Wassum: Senior editor, *eLife*. The other authors declare that no competing interests exist.

### Funding

| Funder | Grant reference number | Author |
| --- | --- | --- |
| National Institutes of Health | DA035443 | Kate M Wassum |
| National Science Foundation | | Ana C Sias |

The funders had no role in study design, data collection and interpretation, or the decision to submit the work for publication.

### Author contributions

Ana C Sias, Conceptualization, Data curation, Software, Formal analysis, Funding acquisition, Validation, Investigation, Visualization, Methodology, Writing - original draft, Writing - review and editing; Ashleigh K Morse, Conceptualization, Data curation, Formal analysis, Investigation, Visualization, Methodology, Writing - review and editing; Sherry Wang, Venuz Y Greenfield, Caitlin M Goodpaster, Tyler M Wrenn, Investigation; Andrew M Wikenheiser, Software, Methodology, Writing - review and editing; Sandra M Holley, Carlos Cepeda, Validation, Investigation, Visualization, Writing - review and editing; Michael S Levine, Conceptualization, Validation; Kate M Wassum, Conceptualization, Resources, Data curation, Formal analysis, Supervision, Funding acquisition, Investigation, Visualization, Methodology, Writing - original draft, Project administration, Writing - review and editing

### Author ORCIDs

Ashleigh K Morse http://orcid.org/0000-0003-0773-5790
Caitlin M Goodpaster http://orcid.org/0000-0002-2456-9010
Kate M Wassum https://orcid.org/0000-0002-2635-7433

### Ethics

Animal experimentation: All procedures were conducted in accordance with the NIH Guide for the Care and Use of Laboratory Animals and were approved by the UCLA Institutional Animal Care and Use Committee.

**Decision letter and Author response**

Decision letter https://doi.org/10.7554/eLife.68617.sa1

Author response https://doi.org/10.7554/eLife.68617.sa2

# Additional files

### Supplementary files

• Source code 1. Source code for fiber photometry data analysis in *Figures 1* and *2*.

• Transparent reporting form

### Data availability

All data and code support the findings of this study are available from the corresponding author upon request and via Dryad (https://doi.org/10.5068/D1109S).

The following dataset was generated:

| Author(s) | Year | Dataset title | Dataset URL | Database and Identifier |
|---|---|---|---|---|
| Wassum KM | 2021 | A bidirectional corticoamygdala circuit for the encoding and retrieval of detailed reward memories | https://doi.org/10.5068/D1109S | Dryad Digital Repository, 10.5068/D1109S |

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
