## [Decision Letter]

**Acceptance summary:**

This study examined the neural mechanism underlying stimulus-outcome associations. Using a series of sophisticated experiments with optogenetics and pharmacogenetics, the authors show that interactions between the basolateral amygdala (BLA) and the lateral part of the orbitofrontal cortex (lOFC) play a critical role in learning to predict the identity of outcome from a cue, but not in valuing valence of outcomes. This advances our understanding of the roles that lOFC and BLA play in associative learning.

**Decision letter after peer review:**

Thank you for submitting your article "A bidirectional corticoamygdala circuit for the encoding and retrieval of detailed reward memories" for consideration by *eLife*. Your article has been reviewed by 3 peer reviewers, including Naoshige Uchida as the Reviewing Editor and Reviewer #1, and the evaluation has been overseen by John Huguenard as the Senior Editor.

Essential revisions:

This study examined the neural mechanism underlying stimulus-outcome associations. Using a series of sophisticated experiments with otpogenetics and pharmacogenetics, the authors show that interactions between the basolateral amygdala (BLA) and the lateral part of the orbitofrontal cortex (lOFC) play critical role in learning to predict the identity of outcome predicted by a cue, but not in learning to predict reward generally. These results extend our understanding of how BLA and lOFC regulate associative learning.

As you will see below, all the reviewers agreed that this study addresses an important question, the authors have performed sophisticated and sound experiments, and the manuscript is written clearly. However, they raised various points that need to be clarified or addressed. The reviewers do not think that additional experiments are required, and all of the concerns can be addressed by additional analysis or revising text. In particular, a little more careful discussions on interpretation of results (e.g. whether the same information is transmitted between lOFC and BLA requires neuronal activity measurements) will be useful.

*Reviewer #3 (Recommendations for the authors):*

Do the eYFP and ArchT control groups show equivalent baseline levels of outcome-specific lever pressing in the PIT procedure absent the Pavlovian cues? There seems to be a lot more variability in the ArchT or ArchT/hM4Di groups potentially reflecting an overall increase or decrease in the amount of instrumental conditioning leading up to the PIT procedure? Particularly in the last experiment the serial disconnections appear to cause a general enhancement of PIT.

Are their instances in which the animals attempt to retrieve rewards during the cue presentation when no reward has been delivered? How are these dealt with in the GCaMP analyses and was optogenetic inhibition delivered surrounding these events? If not, it seems like these "false alarms" might function to degrade sensory-specific reward memories on their own and the optogenetic inhibition might hasten the effect of these errors. This leads into a broader question of what precisely do the authors believe the inhibition surrounding reward retrieval events is doing to the sensory-specific memory? Is it being prevented from forming entirely because the inhibition is done in all eight sessions? Is it simply weakening it? Saying it affects encoding is accurate but a bit vague.

The authors tested for unilateral, ipsilateral effects of the optogenetic or chemogenetic manipulations in the same animals in which the full serial disconnection approach was tested. The unilateral manipulations did not have an effect on PIT performance, but I would like to see that data plotted separately for the unilateral groups broken out by unilateral manipulations of the BLA inputs to lOFC and vice versa, rather than collapsing across the ArchT and hM4Di manipulations. Also, a stronger test of their hypothesis would be to contrast the unilateral performance to the serial disconnection performance (especially when the unilateral is aggregated). However, in comparing Figure 5 and Figure 5-2 it would appear that the main effect of the serial disconnection is not to impede an enhancement of outcome-specific PIT (elevated lever pressing compared to baseline for the unilateral, ipsilateral and bilateral, contralateral manipulations when the CS and lever associations are the same appears intact), but rather to disrupt outcome-specific PIT when there is a mismatch between the CS and the lever reward associations. This appears very different from the pattern of effect from bilateral inhibition of lOFC inputs to BLA, which argues against the hypothesis that the same information is being coded within each part of the circuitry.

---

## [Author Response]

Essential revisions:[…] As you will see below, all the reviewers agreed that this study addresses an important question, the authors have performed sophisticated and sound experiments, and the manuscript is written clearly. However, they raised various points that need to be clarified or addressed. The reviewers do not think that additional experiments are required, and all of the concerns can be addressed by additional analysis or revising text. In particular, a little more careful discussions on interpretation of results (e.g. whether the same information is transmitted between lOFC and BLA requires neuronal activity measurements) will be useful.

We appreciate the reviewers pointing out how our language led to an interpretation that we agree is not supported by the current data. Indeed, the data do not show whether the same information is transmitted between lOFC→BLA and BLA→lOFC and that need not be the case for these projections to function in a circuit. To remedy this, we have removed the ‘same information’ language throughout the manuscript, including in the abstract (Pg. 2), results (Pg. 9-11), discussion (Pg. 13-14), and methods (Pg. 20-21). We have brought our framing and interpretation of the disconnection results much closer to the present data. For example:

Results Pg. 10: “Therefore, we next asked whether the lOFC→BLA and BLA→lOFC pathways form a functional stimulus-outcome memory encoding and retrieval circuit, i.e., whether the sensory-specific associative information that requires lOFC→BLA projections to be encoded also requires activation of BLA→lOFC projections to be used to guide decision making, or whether these are independent, parallel pathways, tapping into essential but independent streams of information.”

Results Pg. 11: “…indicating that the lOFC and BLA form a bidirectional circuit for the encoding (lOFC→BLA) and use (BLA→lOFC) of appetitive stimulus-outcome memories.”

Discussion Pg. 14: “Here, using a serial disconnection procedure, we found that during reward choice BLA→lOFC projection activity mediates the use of the sensory-specific associative information that is learned via activation of lOFC→BLA projections. Thus, lOFC→BLA→lOFC is a functional circuit for the encoding (lOFC→BLA) and subsequent use (BLA→lOFC) of sensory-specific reward memories to inform decision making.”

We have also included the important caveat that future work with detailed characterization of the activity of each pathway is needed to draw conclusions on the information content conveyed by each pathway:

Discussion Pg. 14: *“*The precise information content conveyed by each component of the lOFC-BLA circuit and how it is used in the receiving structure is a critical follow-up question that will require a cellular resolution investigation of the activity of each pathway.”

Reviewer #3 (Recommendations for the authors):The conditioned approach responses appear to asymptote after two out of the eight Pavlovian conditioning sessions. Although the authors have run a control experiment in which they show that novelty contributes to the GCaMP responses measured in BLA at cue onset in early sessions, they do not clearly demonstrate learning related changes in GCaMP responses across sessions to either cue or reward retrieval. Thus, it isn't necessarily clear how quickly the sensory-specific reward memories are formed in BLA and if repeated stimulus-outcome pairings, particularly once general approach behavior reaches asymptote, actually serve to increasingly strengthen the memory.

We agree with this limitation that our report and are actively working to address these interesting questions in our ongoing work. Indeed, a learning related-change in the BLA response can only be inferred from the present data and is not directly demonstrated. In the present experiment the nature of the memory is tested after learning, precluding understanding of the precise time course of the development of the sensory-specific stimulus-outcome memory. Future work should incorporate an online neural and/or behavioral assessment of sensory-specific reward memory encoding during learning to well address this important question.

No explanation is provided for how the transient BLA GCaMP responses at cue onset sustain stimulus-outcome memory encoding at the time of reward. A straightforward account would be a sustained response to the cue that overlaps with the GCaMP response to reward retrieval. In addition there is no attempt to transiently inactivate the entire BLA or specific pathways at cue onset to determine how simple cue encoding affects subsequent performance in the PIT paradigm.

This is an excellent point. We were somewhat surprised to see only a transient response to the CS onset. This suggests to us that perhaps there is a more sustained response elsewhere in the brain (or even in a different cell type in the BLA). Perhaps this sustained response follows the transient response detected here.

We also agree that it is an important question (and limitation of the current work) of whether the BLA response to the cue is important for S-O memories. This is also a question we are addressing on in our ongoing work.

We have acknowledged both this limitation/interesting question in the revised manuscript.

Discussion Pg. 13: “Future work is needed to reveal the precise information content encoded by BLA neurons during reward experience that confers their function in the formation of stimulus-outcome memories, though BLA neurons will respond selectively to unique food rewards (Liu et al., 2018), which could support the generation of sensory-specific reward memories. Whether BLA cue responses are also important for encoding stimulus-outcome memories is another important question exposed by the current results.”

The multiplexed chemogenetic and optogenetic serial disconnection approach is too coarse a manipulation to support the claim that reciprocal connections between the BLA and lOFC support encoding and retrieval of the same information. To make this claim it is necessary to use detailed functional assays of the activity in each pathway to determine what information they code during the Pavlovian conditioning and PIT procedures.

We completely agree with this excellent point. We appreciate the reviewer pointing out how our language led to an interpretation that is not supported by the current data. Indeed, the data do not show whether the same information is transmitted between lOFC→BLA and BLA→lOFC and that need not be the case for these projections to function in a circuit. To remedy this, we have removed the ‘same information’ language throughout the manuscript, including in the abstract (Pg. 2), results (Pg. 9-11), discussion (Pg. 13-14), and methods (Pg. 20-21). We have brought our framing and interpretation of the disconnection results much closer to the present data. For example:

Results Pg. 10: “Therefore, we next asked whether the lOFC→BLA and BLA→lOFC pathways form a functional stimulus-outcome memory encoding and retrieval circuit, i.e., whether the sensory-specific associative information that requires lOFC→BLA projections to be encoded also requires activation of BLA→lOFC projections to be used to guide decision making, or whether these are independent, parallel pathways, tapping into essential but independent streams of information.”

Results Pg. 11: “…indicating that the lOFC and BLA form a bidirectional circuit for the encoding (lOFC→BLA) and use (BLA→lOFC) of appetitive stimulus-outcome memories.”

Discussion Pg. 14: “Here, using a serial disconnection procedure, we found that during reward choice BLA→lOFC projection activity mediates the use of the sensory-specific associative information that is learned via activation of lOFC→BLA projections. Thus, lOFC→BLA→lOFC is a functional circuit for the encoding (lOFC→BLA) and subsequent use (BLA→lOFC) of sensory-specific reward memories to inform decision making.”

We have also included the important caveat that future work with detailed characterization of the activity of each pathway is needed to draw conclusions on the information content conveyed by each pathway:

Discussion Pg. 14: “The precise information content conveyed by each component of the lOFC-BLA circuit and how it is used in the receiving structure is a critical follow-up question that will require a cellular resolution investigation of the activity of each pathway.”

Do the eYFP and ArchT control groups show equivalent baseline levels of outcome-specific lever pressing in the PIT procedure absent the Pavlovian cues? There seems to be a lot more variability in the ArchT or ArchT/hM4Di groups potentially reflecting an overall increase or decrease in the amount of instrumental conditioning leading up to the PIT procedure? Particularly in the last experiment the serial disconnections appear to cause a general enhancement of PIT.

Across all the experiments, there are no significant group differences in lever pressing in the absence of the cues. We had previously shown that there were no differences in instrumental conditioning leading up to PIT test between groups for each experiment (Figures 3-2b, 4-2b, 5-2b). In the revised manuscript, we now also include an analysis showing no group differences in same v. different lever pressing during the baseline periods of the PIT test:

Methods Pg. 24: “In all cases, there were no significant differences in baseline presses between levers in the absence of the CSs (Lever: lowest P = 0.33, F_1,14_ = 1.02), and no effect of group on baseline lever pressing (Group: lowest P = 0.54, F_2,23_ = 0.63; Group x Lever lowest P = 0.21, F_1,14_ = 1.71).”

Are their instances in which the animals attempt to retrieve rewards during the cue presentation when no reward has been delivered? How are these dealt with in the GCaMP analyses and was optogenetic inhibition delivered surrounding these events? If not, it seems like these "false alarms" might function to degrade sensory-specific reward memories on their own and the optogenetic inhibition might hasten the effect of these errors.

Yes, such instances are the majority of the entry responses during the cues and are how we quantified general approach behavior during the conditioning. All of the entry responses we show during the Pavlovian training (Figure 1h, 3e, 4e, 5e) were responses after the cue onset, prior to delivery of the first reward (‘CS probe period’) and, thus, by definition the reward was not present. We have tried to make this clearer in the manuscript:

Figure legends: “Elevation [(CS probe entry rate)/(CS probe entry rate + preCS entry rate)] in food-port entries during the CS probe period (after CS onset, before first reward delivery)…”

Results Pg. 5: “Rats demonstrated simple Pavlovian conditioning by gradually increasing their goal approach responses (entries into the food-delivery port) during the cues (after cue onset, before reward delivery) across training…”

Results Pg. 7: “Optical inhibition of BLA neurons at reward experience during Pavlovian conditioning did not impede the development of the Pavlovian conditional goal-approach response to the cue sampled prior to reward delivery…”

For the GCaMP analyses, to look for a BLA response to reward we focused only on the first food-port entry following a reward delivery, i.e., ‘reward retrieval’. Thus, the data shown in Figure 1g are not impacted by food-port entries in which a reward was absent. We have clarified this throughout the manuscript:

Legend for Figure 1: “(e) Representative examples of GCaMP6f fluorescence changes (Z-scored ∆F/F) in response to CS presentation (blue box), reward delivery (orange tick), and reward retrieval (orange triangle; first food-port entry following reward delivery) across days of training.”

Results Pg. 5: “BLA neurons were robustly activated by both cue onset and reward retrieval (first food-port entry after reward delivery) throughout Pavlovian conditioning.”

Methods Pg. 24: “Using a custom MATLAB workflow, Z-scored traces were then aligned to CS onset, reward delivery, reward retrieval (first food-port entry after reward delivery), and food-port entries without reward present during the CS probe period (after CS before first reward delivery) during the CS for each trial.”

Entries into the food-delivery port when a reward was not present were not accompanied by a BLA response. We now show these data in Figure 1—figure supplement 5 (Pg. 36). Thus, BLA responses to reward retrieval are due to the presence of reward, not the entry response itself. These data are reference in the main text:

Results Pg. 5: “There were no significant BLA activity changes detected in response to food-port entries absent reward (Figure 1-5), indicating that reward retrieval responses resulted from reward experience rather than the act of entering the food port.”

Optical inhibition was not given during entries without a reward present. See the following text from the methods on Pg. 21:

“Light was delivered continuously for 5 seconds concurrent with each reward delivery. If the reward was retrieved (first food-port entry post-delivery) while the light was still being delivered (i.e., within 5 s of reward delivery), then the light delivery was extended to 5 s post retrieval. If the reward was retrieved after the laser had gone off, then it triggered an additional 5 s continuous illumination.”

We don’t think ‘False alarms’ can well characterize entries in this task. During the cues the associated reward is delivered on a random time 30s schedule, so on average every 30s. Thus, the animals cannot predict when during the cue the reward will come and it is not possible for them to have a specific expectation of when during the cue the reward will be present. Therefore, it is not possible to have a false alarm in such a prediction. We think these entries are better characterized as anticipatory conditional responses. Because the animals are not able to precisely time when the reward will be delivered, they are encouraged to regularly check the magazine, and show quite a bit of these Pavlovian conditional goal-approach responses. Such responses may not always completely reflect an expectation of a specific reward or understanding of the cue-reward contingency (e.g., they are not always completely sensitive to outcome devaluation or degradation of the Pavlovian contingency). Given this, it is difficult to know whether the probe period entries actually degrade the sensory-specific reward memory and thus whether the optical stimulation paired with the reward hastens such an effect. Because there are so many of such entries (20+/min) and sensory-specific reward memories are well developed and intact in control subjects (in our experiments and in the many others from our and other labs using this same task), we think it somewhat unlikely that these entries have a major effect to degrade the sensory-specific reward memory. We think a more straightforward explanation of the optogenetic manipulation results is that inhibiting the BLA or lOFC→BLA projections at the time of stimulus-outcome pairing (reward experience during the cue) attenuated the animal’s ability to link the specific reward event to the associated cue.

This leads into a broader question of what precisely do the authors believe the inhibition surrounding reward retrieval events is doing to the sensory-specific memory? Is it being prevented from forming entirely because the inhibition is done in all eight sessions? Is it simply weakening it? Saying it affects encoding is accurate but a bit vague.

Thank you for this thoughtful comment. It prompted us to revaluate our interpretation. First, we think that it is a bit too strong to say that the inhibition in any of the experiments entirely prevented the encoding of stimulus-outcome memories, thus we have changed to the term ‘prevent’ to ‘attenuated’ throughout. We think that inhibiting the BLA or lOFC→BLA projections at the time of stimulus-outcome pairing (reward experience during the cue) attenuated the animal’s ability to link the specific reward event to the associated cue. Whether the memory is absent, weakened, or changed is an interesting question that based on the current data alone we are unable to speculate on. We can say that the encoding of the sensory-specific stimulus-outcome memory is disrupted to the extent that animals are unable to later use those memories to guide choice behavior. We’ve tried to clarify this throughout the text, for example:

Discussion Pg. 13: “Leveraging the temporal resolution of optogenetics, we demonstrated that BLA principal neurons mediate the encoding of such memories, and specifically that activity at the time of reward experience during a cue is critical. […] Whether BLA cue responses are also important for encoding stimulus-outcome memories is another important question exposed by the current results.”

Discussion Pg. 14: “Thus, lOFC→BLA projections may be responsible for linking states, defined by internal physiological and external predictive cues, to the specific rewarding events with which they are associated.”

The authors tested for unilateral, ipsilateral effects of the optogenetic or chemogenetic manipulations in the same animals in which the full serial disconnection approach was tested. The unilateral manipulations did not have an effect on PIT performance, but I would like to see that data plotted separately for the unilateral groups broken out by unilateral manipulations of the BLA inputs to lOFC and vice versa, rather than collapsing across the ArchT and hM4Di manipulations. Also, a stronger test of their hypothesis would be to contrast the unilateral performance to the serial disconnection performance (especially when the unilateral is aggregated).

We apologize for the confusion. The ipsilateral control was conducted in a separate group of subjects, not the same animals in which the full serial disconnection approach was tested. This group received unilateral ipsilateral inhibition of lOFC→BLA projections during learning *and* BLA→lOFC projections during retrieval. We have now clarified this:

Results: Pg 10: “To control for unilateral inhibition of each pathway without disconnecting the circuit, a second control group (N = 8) received the same procedures as the experimental contralateral ArchT/hM4Di disconnection group, except with BLA hM4Di and the lOFC guide cannula in the same hemisphere as the optical fiber used to inactivate lOFC axons and terminals in the BLA (Figure 5-1). Thus, during the PIT test for this group the BLA→lOFC pathway was chemogenetically inactivated in the same hemisphere in which the lOFC→BLA pathway had been optically inactivated during Pavlovian conditioning, leaving the entire circuit undisrupted in the other hemisphere.”

As was also requested by Reviewer #1, we have now incorporated the ipsilateral control group data into the main Figure 5 (Pg. 11). As you can see below, because there were no differences between the two control groups, we combined them into a single control group for comparison to the disconnection group. The individual data points in Figure 5 are coded by control group (eYFP/mCherry solid lines and circles, ipsilateral ArchT/hM4Di dashed lines and triangles). We also provide the disaggregated data showing a comparison between all three groups (contralateral eYFP/mCherry, Ipsilateral inhibition, Contralateral disconnection) in Figure 5-2 (Pg. 43).

Results on Pg. 10: “The control group received identical procedures with the exception that viruses lacked ArchT and hM4Di (N = 8). […] These control groups did not differ on any measure and so were collapsed into a single control group [(Pavlovian training, Training: F_(2.2,31.3)_ = 12.96, P < 0.0001; Control group type: F_(1,14)_ = 0.02, P = 0.89; Group x Training: F_(7.98)_ = 0.76, P = 0.62) (PIT Lever presses, Lever: F_(1,14)_ = 14.68, P = 0.002; Control group type: F_(1,14)_ = 0.38, P = 0.55; Group x Lever: F_(1,14)_ = 0.43, P = 0.52) (PIT Food-port entries, t_14_ = 0.72, P = 0.48)]. See also Figure 5-2 for disaggregated control data.”

The new analyses are now reported in the results Pg. 11: “As with the bilateral inhibition experiments, the control and disconnection groups developed a Pavlovian conditional goal-approach response with training (Figure 5e; Training: F_(2.8,68.1)_ = 28.13, P < 0.0001; Group (Combined control group v. Contralateral ArchT/hM4Di- disconnection): F_(1,24)_ = 0.46, P = 0.51; Group x Training: F_(7,168)_ = 0.44, P = 0.88; see also Figure 5-2a), which was similarly expressed during the PIT test (Figure 5g; t_24_ = 0.11, P = 0.91; see also Figure 5-2d).”

However, in comparing Figure 5 and Figure 5-2 it would appear that the main effect of the serial disconnection is not to impede an enhancement of outcome-specific PIT (elevated lever pressing compared to baseline for the unilateral, ipsilateral and bilateral, contralateral manipulations when the CS and lever associations are the same appears intact), but rather to disrupt outcome-specific PIT when there is a mismatch between the CS and the lever reward associations. This appears very different from the pattern of effect from bilateral inhibition of lOFC inputs to BLA, which argues against the hypothesis that the same information is being coded within each part of the circuitry.

This is an excellent point and we appreciate the suggestion to discuss it. Indeed, following serial disconnection of lOFC→BLA projections during learning from BLA→lOFC projections during retrieval we find that the outcome-specificity of the PIT effect is disrupted, but, unlike inhibition of the BLA itself or lOFC→BLA projections, there is a non-discriminate excitation of instrumental behavior. That is, during the cue the rats increase pressing on both levers. We now make this clear in the results.

Results Pg. 11: “Whereas in the control group cue presentation significantly biased choice towards the action earning the same predicted reward, this outcome-specific PIT effect did not occur in the disconnection group. Rather, during the cues rats in the disconnection group showed a non-discriminate elevation in pressing on both levers (Figure 5-2c).”

We also now discuss this interesting finding in the discussion. We completely agree that this suggests that the same information is *not* being coded in the lOFC→BLA and BLA→lOFC pathways, and regret that our prior language gave that impression. For lOFC→BLA→lOFC to be a functional stimulus-outcome memory circuit it need not be the case that the same information is coded in each part of the pathway. Generally, we think this result suggests that the BLA itself and lOFC→BLA projections mediates the learning of a broader set of information than that being transmitted back to the lOFC by BLA→lOFC projections. Some of that information, which would have been capable of being encoded in the hemisphere without lOFC→BLA projection inactivation in the disconnection group, could be capable of promoting reward-seeking activity more broadly. Thus, whereas lOFC→BLA projections facilitate the encoding of many aspects of the reward memory, BLA→lOFC projections are only needed to access the subset of sensory-specific features needed to allow animals to know during the cue which specific reward is predicted and thus which action to select. See our revised discussion.

Discussion, Pg. 14: “Here, using a serial disconnection procedure, we found that during reward choice BLA→lOFC projection activity mediates the use of the sensory-specific associative information that is learned via activation of lOFC→BLA projections. […] Indeed, BLA→lOFC are not the only amygdala projections involved in reward memory (Beyeler et al., 2016; Corbit et al., 2013; Fisher et al., 2020; Kochli et al., 2020; Morse et al., 2020; Parkes and Balleine, 2013).”